

# Development of an inorganic and organic aerosol model (Chimere2017β v1.0): seasonal and spatial evaluation over Europe

Florian Couvidat[1], Bertrand Bessagnet[1], Marta Garcia-Vivanco[2], Elsa Real[1], Laurent Menut[3], and Augustin Colette[1]

[1]Institut National de l'Environnement Industriel et des Risques, Verneuil-en-Halatte, France
[2]Centro de Investigaciones Energéticas, Medioambientales y Tecnológicas (CIEMAT), Departamento de Medio Ambiente, Av. Complutense 22, 28040-Madrid, Spain
[3]Laboratoire de Métérologie Dynamique, IPSL, CNRS, UMR8539, 91128 Palaiseau Cedex, France

*Correspondence to:* Florian Couvidat
(Florian.Couvidat@ineris.fr)

**Abstract.**

A new aerosol module was developed and integrated in the air quality model CHIMERE. Developments include an update of biogenic emissions and of the inorganic thermodynamic model ISORROPIA, revision of wet deposition processes and of the algorithms of condensation/evaporation and coagulation and the implementation of the SOA mechanism H$^2$O and the thermodynamic model SOAP.

Concentrations of particles over Europe were simulated by the model for the year 2013. Model concentrations were compared to the EMEP program observations and other observations available in the EBAS database to evaluate the performances of the model. Performances were determined for several components of particles (sea salts, sulfate, ammonium, nitrate, organic aerosol) with a seasonal and regional analysis of results.

The model gives good performances in general. For sea salts, the model succeeds in reproducing the seasonal evolution of concentrations for Western and Central Europe. For sulfate, except for an overestimation of sulfate in Northern Europe, modeled concentrations are close to observations with a good seasonal evolution of concentrations. For organic aerosol, the model performs well for stations with strong modeled biogenic SOA concentrations.

However, the model strongly overestimates ammonium nitrate concentrations during late autumn (possibly due to problems in the temporal evolution of emissions) and strongly underestimates SOA concentrations over most of stations (especially in the Northern half of Europe). This underestimation could be due to a lack of anthropogenic SOA in the model.

A list of recommended tests and developments to improve the model is also given.

## 1 Introduction

Atmospheric Particulate Matter (PM) contributes to adverse effects on health and ecosystems. The development of models is necessary to predict the formation of particles in order to evaluate their concentrations and to evaluate mitigation strategies. However, developing models with enough precision is quite challenging due to the complexity and variety of phenomena and



the great number of chemical species involved. Numerous air quality models have been developed to simulate PM concentrations (Emmons et al., 2010; Pozzoli et al., 2011; Zhang et al., 2010; Vestreng, 2003; Carlton et al., 2010; Menut et al., 2013; Sartelet et al., 2007).

PM is constituted of various chemical species: organic matter (OM), elemental carbon (EC) mainly originating from an-
thropogenic sources, major inorganic components (ammonium, nitrate and sulfate), sea salt, mineral dust and other crustal compounds. These species originate from numerous emission sources which can be natural (biogenic emissions from vegetation, sea-salt emissions, dust emissions) or anthropogenic (for example emissions from residential biomass burning, road traffic, agriculture, industrial sources). Particles can be primarily emitted in the atmosphere or secondary formed from chemical reactions.

OM typically represents between 20 and 60% (Kanakidou et al., 2005; Yu et al., 2007; Zhang et al., 2007) of the fine particulate mass and is formed via the partitioning of Semi-Volatile Organic Compounds (SVOC) between the gas and particle phases. These SVOCs can be primary in origin but OM is often considered to be mainly constituted of secondary organic compounds formed via the oxidation in the atmosphere of Volatile Organic Compounds (VOC) which can be biogenic (like isoprene, monoterpenes and sesquiterpenes) or anthropogenic (for example long-chain alkanes, toluene and other aromatics).

The oxidation (in the gas phase or in the aqueous phase) of sulfur dioxide $SO_2$ produces sulfuric acid $H_2SO_4$ which leads to sulfate formation via condensation or nucleation processes. If ammonia $NH_3$ is present in the atmosphere, it will neutralize sulfate and form ammonium. If there is still $NH_3$ available in the gas phase, it can lead to the formation of ammonium nitrate in the presence of nitric acid $HNO_3$ (formed via the oxidation of nitrogen oxides $NO_x$). Sea salts and natural dusts are primary natural particles and can interact with atmospheric pollutants. For example, $HNO_3$ can condense onto sea salts and leads to
sodium nitrate and then to the volatilization of chloride acid (HCl). Similarly, $HNO_3$ can condense onto dust and leads to the formation of calcium nitrate. As dust and sea salts are mainly coarse particles, these two processes can lead to the formation of coarse nitrate whereas ammonium nitrate will mainly remain in fine particles.

To simulate PM concentrations, models have to take into account the microphysics of particles (condensation/evaporation, coagulation, nucleation), chemical mechanisms for the gas-phase chemistry, aerosols thermodynamics, emissions and deposi-
tion processes. In the scope of this study, a new aerosol module has been developed in a modified version of the CHIMERE model. These modifications include an update of biogenic emissions and microphysics parameterizations, the implementation of thermodynamics model ISORROPIA v2.1 (Fountoukis and Nenes, 2007) and SOAP (Couvidat and Sartelet, 2015), the SOA mechanism of Couvidat et al. (2012) and modifications of deposition parameterizations. The results of the model were evaluated by comparison to measurements of $PM_{2.5}$ and $PM_{10}$ concentrations but also of composition (Cl, Na, $SO_4$, $NO_3$, $NH_4$,
organic carbon).

The representation of several processes was revised to improve the results of the model:

- Biogenic emissions are computed with the MEGAN 2.1 algorithm (Guenther et al., 2012) with updated emission factors and Leaf Area Index (LAI) data.



- Below-cloud scavenging is represented as in Henzing et al. (2006) with a polydispersed distribution of cloud droplets (providing a distribution of droplet diameter as a function of rainfall). In-cloud scavenging is represented with the algorithm of Croft et al. (2010).

- Evaporation/condensation of semi-volatile species is represented with the algorithm of Pandis et al. (1993) using thermodynamic equilibria. Coagulation of particles is represented as in Debry et al. (2007). Thermodynamic equilibria are computed with the ISORROPIA II model (Fountoukis and Nenes, 2007) for inorganic compounds and with the Secondary Organic Aerosol Processor SOAP (Couvidat and Sartelet, 2015) for organic compounds. The CHIMERE 2013 version used an older version of ISORROPIA 2.1 and a model for organic aerosol based on Pun et al. (2002).

- The SOA formation mechanism of Couvidat et al. (2012) is used for toluene, xylene and biogenic VOC. The CHIMERE 2013 version used the mechanism of Bessagnet et al. (2008).

- The amount of water in particles is calculated as a function of humidity and the composition of particles using ISOR-ROPIA. This amount is used to calculate the wet density of particles (with water) and the wet diameter of particles which are used to compute the kinetics of absorption, coagulation and deposition instead of the dry values as in Menut et al. (2013).

The aerosol module is described in the part of the paper. The second part focuses on the comparison of modeled concentrations with observations for Cl, Na, $SO_4$, $NO_3$, $NH_4$, organic carbon, $PM_{2.5}$ and $PM_{10}$ with a regional and seasonal analyses of results.

## 2 Method

The CHIMERE 2013 version (Menut et al., 2013) was modified to update some parameterizations and chemical mechanisms. The modified version of CHIMERE was then evaluated for the simulation of PM concentration and composition over Europe in 2013.

### 2.1 Model development

#### 2.1.1 Chemical mechanisms

A simple aqueous-phase chemical mechanism is used for sulfate formation from the oxidation of $SO_2$ in clouds. This mechanism assumes that the aqueous-phase concentrations of $SO_2$, $H_2O_2$ and $O_3$ are at equilibrium with the gas phase with a partitioning being function of pH for $SO_2$. The pH of clouds is computed by taking into account the absorption and dissociation of various acids ($H_2SO_4$, HCl, $HNO_3$ and $H_2CO_3$) and the formation of $NH_4^+$. The electroneutrality equation is solved with the Newton-Raphson method. Henry's law constants, equilibrium constants are taken from Seinfeld and Pandis (1998).

For SVOC formation that leads to the formation of SOA compounds after partitioning, the mechanism of Couvidat et al. (2012) is used. The mechanism is shown in Table 1. It takes into account the formation of SVOC from biogenic (isoprene,





monoterpenes, sesquiterpenes) and anthropogenic precursors (toluene, xylenes) under high-$NO_x$ and low-$NO_x$ conditions.

In Couvidat et al. (2012), Primary Organic Aerosols (POA) are assumed to be SVOC and are split into three compounds POAlP, POAmP and POAhP (having respectively a low, medium and high volatility) to follow the dilution curve of POA in Robinson et al. (2007). The aging of these compounds is also taken into account with a reaction with OH which leads to less

volatile compounds SOAlP, SOAmP and SOAhP via the following reactions:

$$POAlP + OH \xrightarrow{k} SOAlP \tag{1}$$

$$POAmP + OH \xrightarrow{k} SOAmP \tag{2}$$

$$POAhP + OH \xrightarrow{k} SOAhP \tag{3}$$

with k the kinetic rate constant equal to $2 \times 10^{-11}$ molecules$^{-1}$.cm$^3$.s$^{-1}$. Following Grieshop et al. (2009), aging was assumed to lead to a decrease of volatility by a factor 100 (SOAlP, SOAmP and SOAhP are respectively less volatile by a factor 100

than POAlP, POAmP and POAhP).

### 2.1.2 Biogenic emissions

Biogenic emissions are computed with the Model of Emissions and Gases and Aerosols from Nature MEGAN 2.1 algorithm (Guenther et al., 2012), which was implemented in CHIMERE. It uses meteorological conditions (Temperature, Solar radiation

and soil moisture), the Leaf Area Index and the Plant Functional Type (PFT) to compute biogenic emissions. In this study, the above-canopy model is used. The effect of soil moisture on isoprene emissions is not taken into account because of no Wilting Point (i.e. the soil moisture level below which plants cannot extract water from soil) database are available over Europe. Therefore, isoprene emissions may be overestimated during dry periods.

High spatiotemporal data (30 arc-seconds every 8 days) generated from MODIS (Yuan et al., 2011) were used for LAI

inputs. The 30 arc-seconds USGS (US Geophysical Survey) landuse database was used to provide information on the plant functional type. The PFT was then combined with the emissions factors for each functional type of Guenther et al. (2012) to compute the landscape average emissions factors.

### 2.1.3 Anthropogenic emissions

VOC emissions are used as in Menut et al. (2013) except for primary SVOC emissions, a SVOC/POA factor is applied to convert POA emissions into SVOC emissions. In Couvidat et al. (2012), a SVOC/POA factor of 5 was used on the basis that SVOC primary emissions were underestimated. With this factor, the model was able to simulate the strong concentrations of



organic aerosols in winter and to give satisfactory results over most of Europe. Denier van der Gon et al. (2015) has shown that POA emissions are greatly underestimated due to a strong underestimation of residential wood burning emissions by a factor 3 over Europe (between 1 and 10 depending on the countries) if SVOC emissions are included. By correcting POA emissions and assuming that Intermediate-Volatility Organic Compounds (IVOC) are missing from the inventory (with the assumption

that IVOC emissions are equal to 1.5 POA emissions), the authors obtained satisfactory results in winter but still with an underestimation of OM from biomass burning. The authors used therefore a total (IVOC+SVOC)/POA factor of 7.5.

In this study, a SVOC/POA of 5 is used for residential emissions (without adding IVOC emissions) and 1 for other sectors (assuming therefore that no SVOC emissions from other sectors are missing). However, IVOCs are not taken into account because of the large uncertainties on their emissions and their oxidation mechanism. Pye and Seinfeld (2010) used naphthalene

as a surrogate for IVOC and used the yields from smog chamber experiments (Chan et al., 2009; Kautzman et al., 2010) to develop a mechanism of IVOC oxidation. The authors found that only minor concentrations of SOA are formed from IVOC (only 5% of total OM) whereas Zhao et al. (2016a) simulated strong concentrations of SOA from IVOC contributing to half the OA over China. However, Pye and Seinfeld (2010) argued that naphthalene may not be an appropriate choice for the surrogate species. Platt et al. (2013) investigated the SOA formation from gasoline vehicle in an environmental reaction chamber and

found that only a small part of SOA could be explained by the oxidation of aromatics compounds and therefore most of the SOA formation could be attributed to the oxidation of IVOC. This result is however contradicted by Nordin et al. (2013) who found that most of the SOA formation is due to the oxidation of aromatics compounds.

### 2.1.4 Thermodynamic of Secondary organic and inorganic aerosol

Two thermodynamic modules were implemented inside CHIMERE to take into account the formation of secondary aerosols: ISORROPIA v2.1 (Fountoukis and Nenes, 2007) for inorganic aerosols and the Secondary Organic Aerosol Processor (SOAP) (Couvidat and Sartelet, 2015) for organic aerosols. Due to the lack of information on the dust composition, crustal elements are not taken into account for the partitioning whereas it can strongly impact the formation of ammonium nitrate (Ansari and Pandis, 1999; Moya et al., 2002). However, a simple reaction (described in section "Condensation/evaporation") is added to

CHIMERE to take into account the formation of calcium nitrate as done by Hodzic et al. (2006).

SOAP computes the partitioning of organic compounds between the gas and particle phases according to the complexity required by the user. It uses the molecular surrogate approach in which surrogate compounds are associated with molecular structures to estimate several properties and parameters (hygroscopicity, absorption into the aqueous phase of particles, activity coefficients and phase separation). Each surrogate can be hydrophilic (condenses only into the aqueous phase of particles), hy-

drophobic (condenses only into the organic phases of particles) or both. Activity coefficients are computed with the UNIFAC (UNIversal Functional group Activity Coefficient; Fredenslund et al. (1975)) thermodynamic model for short-range interactions and with the Aerosol Inorganic-Organic Mixtures Functional groups Activity Coefficients (AIOMFAC) parameterization for medium- and long-range interactions between electrolytes and organic compounds (Zuend et al., 2008, 2011; Zuend and Seinfeld, 2012; Ganbavale et al., 2015).



SOAP can simulate SOA formation with either an equilibrium representation or a dynamic representation of organic aerosol condensation processes. The dynamic representation takes into account the condensation/evaporation kinetic of organic compounds and their diffusion in the particle by dividing the organic particle into several layers. However, this method requires a lot of computing time. Therefore, in a first approach, the equilibrium approach of SOAP is used.

As in Couvidat et al. (2012), SOA surrogate compounds are assumed to be either hydrophilic or hydrophobic and the impact of medium-range and long-range interactions on activity coefficients are not taken into account. For hydrophilic acids, SOAP takes into accounts for the dissociation of organic acids at high pH as a function of their dissociation constant. Moreover, Pun and Seigneur (2007) developed a parameterization to take into account the impact of pH on the oligomerization of aldehyde compounds by computing an effective Henry's law constant:

$$H_{eff} = H \left( 1 + 0.1 \left( \frac{a(H^+)}{10^{-6}} \right)^{1.91} \right) \tag{4}$$

where $H_{eff}$ is the effective Henry's law constant of BiA0D (surrogate species of the $H_2O$ mechanism for aldehydes formed from the oxidation of monoterpenes), H is the monomer Henry's law constant of BiA0D, and $a(H^+)$ is the activity of protons in the aqueous phase.

Thermodynamic properties of biogenic and anthropogenic species are shown in Tables 2. Table 3 shows the properties of primary SVOC compounds (POAlP, POAmP, POAhP) and their aging products.

### 2.1.5 Computation of the wet diameter and the wet density of particles

Several parameterizations (condensation/evaporation, coagulation, particle deposition) depend on the particle diameter $D_{p,wet}$ which is different from the dry diameter (without water) $D_{p,dry}$. Similarly, dry deposition of particles depends on the wet density $d_w$ of particles.

To compute the wet diameter $D_{p,wet}$ and the wet density $d_w$, ISORROPIA is used to compute the amount of water absorbed by each size bin as a function of the composition and the relative humidity. The method of Semmler et al. (2006) is used to compute the density of the liquid aqueous phase $d_l$. The volume of the whole particle is computed with:

$$V_{tot} = V_{solid} + V_{liq,inorg} + V_{org} \tag{5}$$

With $V_{tot}$ the volume of the whole particle, $V_{solid}$ the volume of the solid part of the particle (including dust, black carbon), $V_{liq,inorg}$ the volume of the aqueous phase (including Na, Cl, $SO_4$, $NH_4$, $NO_3$ and $H_2O$) and $V_{org}$ the volume of the organic phase of particles. For simplification purposes, the organic phase density is assumed to be equal to 1300 kg/m$^3$ and the density of the aqueous phase is assumed to be not influenced by hydrophilic organic compounds.

Using the density of the solid phase $d_{sol}$ (assumed to be equal to 2200 kg/m$^3$), the density of the liquid aqueous phase $d_l$ and the density of the organic phase $d_{org}$, Eq. 5 leads to:

$$d_w = \left( \frac{w_{solid}}{d_{solid}} + \frac{w_{liq,inorg}}{d_l} + \frac{w_{org}}{d_{org}} \right)^{-1} \tag{6}$$





With $w_{solid}$, $w_{liq,inorg}$ and $w_{org}$ the mass fraction of respectively the solid phase, the aqueous phase and the organic phase. The wet diameter can be computed with the following equation given by the ratio of the volume of the wet particle to the volume of the dry particle:

$$\frac{D^3_{p,wet}}{D^3_{p,dry}} = \frac{1}{1 - w_{H_2O}} \frac{d_{dry}}{d_{wet}} \tag{7}$$

with $w_{H_2O}$ the mass fraction of water in the particle and $d_{dry}$ the dry density of the particle which can be computed with Semmler et al. (2006) and Eq. 5 without taking into account the mass of water.

### 2.1.6 Dry deposition of particles and semi-volatile organic species

Dry deposition is parameterized via a downward flux $F_{dry,i}$ such as:

$$F_{d,i} = -v_{d,i} * C_i \tag{8}$$

with $v_d$ the deposition velocity. The deposition velocity is represented via the resistance analogy of Wesely (1989). For each gaseous species $i$, $v_{d,i}$ is calculated with:

$$v_{d,i} = \frac{1}{R_a + R_{b,i} + R_{c,i}} \tag{9}$$

with $R_a$ the aerodynamic resistance associated with turbulent transport in the atmosphere, $R_{b,i}$ the quasi-laminar resistance and

$R_{c,i}$ the surface resistance.

The surface resistance depends on the nature of the surface and is generally divided into three categories: water, ground and vegetation. For the deposition of gases to water and vegetation, the parameterizations depend on the Henry's law constants of compounds i. For $O_3$, $SO_2$, $NO_2$, NO and NH3, the values of Menut et al. (2013) are used.

For SVOC, Bessagnet et al. (2010) showed that without dry deposition of gas-phase SVOC could lead to an overestimation of

SOA by 50 %. As done by Bessagnet et al. (2010), Henry's law constants of SVOC are used to take into account their deposition. The Henry's law constants of hydrophilic species are taken from Couvidat et al. (2012). For hydrophobic species, they were calculated using the activity coefficients at infinite dilution as in Couvidat and Seigneur (2011), by using the saturation vapour pressure $P^0$ and the activity coefficient of compound i at infinite dilution computed with UNIFAC $\gamma_i^\infty$ such as:

$$H_i = \lim_{C_i \to 0} \left( \frac{C_i}{P_i} \right) = \frac{\rho_{eau}}{M_{eau} \times \gamma_i^\infty \times P_i^0} \tag{10}$$

with $\rho$ the density of water. For primary SVOC (POAlP, POAmP and POAhP), a Henry's law constant of 0.01 mol/L/atm is used (similar to the Henr's law constant of alkanes). For their aging products, a Henry's law constant of 3000 mol/L/atm is used (for a slightly oxidized molecule). Table 4 shows the Henry's law constants used in this study.

For dry deposition of particles, the parameterizations of Menut et al. (2013) are used. However, the wet diameter and the wet density are used instead of the dry values.





### 2.1.7 Wet deposition of particles and semi-volatile species

In-cloud scavenging for both gases and aerosols is represented by the parameterization of Croft et al. (2010), assuming that wet deposition by in-cloud scavenging is proportional to the amount of cloud water lost by precipitations, such as:

$$\frac{dC}{dt}\bigg|_{incl} = \frac{\zeta_l f_l p_r}{w_l h} C \tag{11}$$

with $p_r$ the precipitation rate (in $\text{g cm}^{-2}\,\text{s}^{-1}$), $w_l$ the liquid water content of clouds (in $\text{g cm}^{-3}$), C the concentration and h the height of the cell (in cm) $f_l$ is the fraction of the compound present in the cloud and $\zeta_l$ an empirical uptake coefficient chosen equal to 1.

For gases, $f_l$ is computed with the effective Henry's law constant and the liquid water content. For particles, $f_l$ is taken as 1 except for particles with a diameter lower than a dry critical radius (chosen equal to 0.1 $\mu$m) which are assumed to be too small to form clouds due to the Kelvin effect.

For the below-cloud scavenging of gases and particles, deposition is described by a scavenging coefficient $\lambda$ (in $\text{s}^{-1}$) such as:

$$\frac{dC}{dt} = -\lambda C \tag{12}$$

For gases, the scavenging coefficient $\lambda_g$ can be calculated with the following equation assuming an irreversible scavenging (Seinfeld and Pandis, 1998):

$$\lambda_g = \int_0^\infty 2\pi D_{dif} Sh R N(R) dR \tag{13}$$

with R the radius of the droplet colliding with the gas, $D_{dif}$ the molecular diffusion coefficient of the deposited compound, Sh is the Sherwood number describing the transfer of gases from air towards a raindrop and N(R) is the number of rain droplets distribution function.

For particles, the scavenging coefficient is expressed by (Seinfeld and Pandis, 1998):

$$\lambda_g = \int_0^\infty \pi R^2 U_t E(R, R_p) N(R) dR \tag{14}$$

with $U_t$ the terminal velocity of the droplet (in m/s) and $E(R,R_p)$ the collision efficiency between a droplet of radius R and a particle of radius $R_p$.

Following Henzing et al. (2006), the rain droplet velocity parameterization of Mätlzer (2002) and the rain droplet size distribution parameterizations of de Wolf (1999) are used:

$$U_t = 0 \qquad\qquad\qquad R < 0.015 mm \tag{15}$$

$$= 4.323(R - 0.015) \qquad\qquad 0.015 \leq R \leq 0.3 mm \tag{16}$$

$$= 9.65 - 10.3 exp(-0.3R) \qquad\qquad R > 0.3 mm \tag{17}$$





$$N(R) = \left(1.047 - 0.0436 * ln(P) + 0.00734 * (lnP)^2\right) \times 1.98 \times 10^{-5} P^{-0.384} R^{2.93} exp\left(-5.38 P^{-0.186R}\right) \qquad (18)$$

### 2.1.8 Condensation/evaporation

Absorption is described by the "bulk equilibrium" approach of Pandis et al. (1993). In this approach, all the bins for which con-
densation is very fast are merged into a "bulk particulate phase". Following Debry et al. (2007), a cutting diameter of 1.25 $\mu m$
is used to separate bins which are inside the "bulk particle" (with a diameter lower than the cutting diameter). Thermodynamic
models are used to compute the partitioning between the gas and particle phases and estimate the gas-phase concentrations at
equilibrium. The equilibrium concentration $G_{eq}$ is calculated by the thermodynamic module ISORROPIA for inorganic semi-
volatile compounds and by SOAP for SVOC.

The mass of compounds condensing onto particles $\Delta A_p$ is redistributed over bins according to the kinetic of condensation
into each bin whereas the mass of compounds evaporating from each bin is proportional to the amount of the compounds in
the bins. If the variation of particulate bulk concentration of compound i $\Delta A_{p,i} > 0$:

$$\Delta A_{p,i}^{bin} = \frac{k^{bin}}{\sum_j k_i^j} DeltaA_{p,i} \qquad (19)$$

with $k_i^{bin}$ the kinetic of condensation given by Seinfeld and Pandis (1998):

$$k_i^{bin} = Number^{bin} \frac{2\pi D_{p,wet}^{bin} D_i M_i}{RT} f(Kn,\alpha) \qquad (20)$$

with $Number^{bin}$ the number of particles inside the bin, $D_p^{bin}$ the mean diameter of the bin, $D_i$ the diffusion coefficient for
species i in air, $M_i$ its molecular weight and f(Kn,$\alpha$) is the correction due to non-continuum effects and imperfect surface
accommodation.

If the variation of particulate bulk concentration of compound i $\Delta A_{p,i} < 0$:

$$\Delta A_{p,i}^{bin} = \frac{A_{p,i}^{bin}}{\sum_j A_{p,i}^j} DeltaA_{p,i} \qquad (21)$$

The absorption flux J ($\mu g m^{-3} s^{-1}$) of a semi-volatile inorganic or organic species onto a bin computed with:

$$J = \frac{1}{\tau} \Delta A_{p,i}^{bin} \qquad (22)$$

with $\tau$ the time to reach equilibrium (chosen equal to the time step of integration).

The gas to particle conversion of $HNO_3$ onto dust and sea salts is also taken into account. $HNO_3$ can indeed react with
calcite $CaCO_3$ of dusts to form calcium nitrate $Ca(NO_3)_2$. $HNO_3$ can also react with dolomite ($MgCa(CO_3)_2$) but only the
reaction with calcite is taken into account. Formenti et al. (2008) found a mass fraction of Ca in dusts between 4% and 9%. A
calcium fraction of 6% is used. In sea salts, $HNO_3$ can replace the Cl present in sea salt and leads to the volatilization of HCl.
Both reactions were assumed to be limited by the condensation kinetic of $HNO_3$ onto particles as in Hodzic et al. (2006).





### 2.1.9 Coagulation

The flux of coagulation $J^{bin}_{coag,i}$ of a coumpound $i$ inside a bin $b$ is computed with the size binning method of Jacobson and Turco (1994):

$$J^b_{coag,i} = \sum_{j=1}^{b} \sum_{k=1}^{b} f^b_{j,k} K_{j,l} A^j_{p,i} Number^k - A^b_{p,i} \sum K_{bin,j} Number^k \tag{23}$$

with $K_{j,l}$ the coagulation kernel coefficient between bins $i$ and $j$ and $f^b_{j,k}$ the partition coefficient (the fraction of the particle created from the coagulation of bins j and k which is redistributed inside bin b). The coagulation kernel and the partition coefficient are calculated as in Debry et al. (2007).

### 2.2 Simulation set-up

The new model was run to simulate the concentrations of particles and their composition in 2013 over Europe with a resolution of 0.25°x0.25°. Meteorology was obtained from Integrated Forecasting System (IFS) model of the European Centre for Medium-Range Weather Forecasts (ECMWF). The meteorology was evaluated in Bessagnet et al. (2016) for 2-meter Temperature, 10-meter Wind Speed and the Planetary Boundary Layer (PBL) for the model intercomparison project Eurodelta III. The authors reported high correlations for temperature (between 0.88 and 0.94) over the whole domain and a slight underestimation of Temperature (between -0.3K and -0.7K), an overestimation of the Wind Speed from 0.1 to 0.9 m/s and an underestimation of the PBL around -100 m (although ECMWF in the Eurodelta III project was shown to be one of model with the lowest RMSE). Anthropogenic emissions of gases and particles were taken from the EMEP inventory (Vestreng, 2003) and Boundary Conditions were generated from the Model for OZone And Related Tracers (Mozart v4.0 (Emmons et al., 2010)).

For PM$_{2.5}$, PM$_{10}$ and each component of PM, several statistics were computed: Root Mean Square Error (RMSE), the correlation coefficient, the Mean Fractional Error (MFE) and the Mean Fractional Bias (MFB). Boylan and Russell (2006) defined two criteria to estimate the performances of the model. The model performance criteria (described as the level of accuracy that is considered to be acceptable for modeling applications) is reached when MFE $\leq$ 75% and when MFB $\leq \pm$ 50% whereas the performance goal (described as the level of accuracy that is considered to be close to the best a model can be expected to achieve) is reached when MFE $\leq$ 50% and when MFB $\leq \pm$ 30%. Although these criteria are not recent, they provide a useful basis to evaluate models.

The seasonal evolution of statistics was examined to study the performances of the model for different seasons and to separate performances over a month from annual performances. The statistics were also computed by "regions" gathering countries having similar features. 5 regions were selected:

– Southern Europe gathering Spain, Portugal and Italy

– Western Europe gathering Ireland, Great Britain and France





- Central Europe gathering Germany, Belgium, Netherlands, Switzerland, Denmark and Austria

- Northern Europe gathering Norway, Sweden and Finland (characterized by low temperatures and low concentrations of particles)

- Eastern Europe gathering the other countries at the east of Europe

A map of regions and stations referred hereafter is shown in Figure 1.

### 2.3  Observations

Results of the model are compared to various measurements ($NO_3$, $NH_4$, $SO_4$, Na, Cl, OC, $PM_1$, $PM_{2.5}$ and $PM_{10}$) available in the EBAS database. EBAS is a database hosting observation data of atmospheric chemical composition and physical properties in support of a number of national and international programs ranging from monitoring activities to research projects. EBAS is
developed and operated by the Norwegian Institute for Air Research (NILU). This database is mostly populated by the EMEP (European Monitoring and Evaluation Programme) measurements (Tørseth et al., 2012).

## 3  Results

### 3.1  Sea salts

Annual scores for sodium (Na) and chloride (Cl) are given in Table 5. Comparisons are carried out over 38 stations for Na
and 35 stations for Cl. Scores are very similar between Na and Cl. The simulated mean concentrations are close (0.67 $\mu$g m$^{-3}$ for Na and 1.28 $\mu$g m$^{-3}$ for Cl) to the measured mean concentrations (0.69 $\mu$g m$^{-3}$ for Na and 1.17 $\mu$g m$^{-3}$ for Cl) and the spatiotemporal correlations are high (0.66 for Na and 0.67 for Cl). MFB are low (6% for Na and 9% for Cl) and MFE (52% for Na and 49% for Cl) are close to the goal criteria of Boylan and Russell (2006) (MFE $\leq$ 50% and MFB $\leq$ $\pm$ 30%). Figure 2 shows the annual concentrations and MFB of Na and Cl at each station. Only one station in Spain underestimates (along
the Bay of Biscay) concentrations for Na. Most stations in Spain, Central Europe and Western Europe have a low annual bias for Na whereas most stations in Northern and Eastern Europe seem to have a high MFB with overestimated concentrations. Results are similar for Cl except in Spain with overestimations observed near the Mediterranean Sea and in some stations in Central Europe far from the seas. This may indicate that the kinetic of $HNO_3$ condensation onto sea salts and the evaporation of HCl is not important enough.

Figure 3 shows the seasonal evolution of the statistics by regions. The same behavior was found for Cl concentrations than for Na concentrations.

Na concentrations seem to be underestimated for the stations in the Southern Europe (only stations in Spain for Na and Cl) from April to October with MFB reaching -60% and the MFE is between 60% to 80% throughout all the year. The temporal correlation is high but the spatial correlation is low. However, for the station ES0008R along the Bay of Biscay with strong ob-
served of Na (with several peaks higher than 6 $\mu$g m$^{-3}$), the model underestimates the concentrations. It could then be possible





that Na concentrations at this station cannot be reproduced due to the low resolution of the model and the strong evolution of concentrations between the sea and the land. The other stations have all a similar pattern shown in Figure 3. Concentrations of Na are overestimated by the model in late Autumn and Winter (with a MFB from 30% to 60%) whereas concentrations are underestimated from June to October with a MFB of -60%. Measurements give higher concentrations of Na in summer and

lower concentrations in winter whereas the model simulate the opposite trend.

For the stations in Western Europe and Central Europe, the model gives satisfactory results and is able to reproduce the seasonal evolution of Na concentrations with MFE around 40 % and MFB between +20% and -20% except for February in Central Europe. Correlations are high (between 50% and 80% in Western Europe around 80% in Central Europe). RMSE are relatively low for Western Europe (between 0.6 and 1.1 $\mu$g m$^{-3}$ for concentrations between 0.8 and 1.7 $\mu$g m$^{-3}$) whereas

RMSE for Central Europe are of the same range than measured and modeled concentrations (between 0.4 and 1.6 $\mu$g m$^{-3}$).

For Eastern and Northern Europe, the model overestimates concentrations throughout the year with MFB often higher than 50% for Eastern Europe and often higher than 30% for Northern Europe and with high MFE (often higher than 50% and even exceeding 100% for some months in Eastern Europe). However, if relative errors are high in Eastern Europe, absolute errors are low (RMSE lower than 0.35 $\mu$g m$^{-3}$) because concentrations in Eastern Europe are very low (mean concentrations lower

than 0.12 $\mu$g m$^{-3}$ and modeled concentrations between 0.09 and 0.35 $\mu$g m$^{-3}$). This overestimation could be due to a lack of sea salts deposition in the model which becomes significant for low concentrations far from seas. Such an underestimation of deposition was reported in Tsyro et al. (2011); Neumann et al. (2016).

### 3.2  Sulfate

Annual scores for SO$_4$ are given in Table 5. Comparisons are carried out over 56 stations. The simulated mean concentrations (1.66 $\mu$g m$^{-3}$) and the measured mean concentrations (1.60 $\mu$g m$^{-3}$) are very close. The spatiotemporal correlation is high (0.67). MFB is low (13%) but indicates a slight relative overestimation. MFE is below 50% (44%) and therefore the goal criteria of Boylan and Russell (2006) is respected for sulfate. The RMSE is equal to 1.13 $\mu$g m$^{-3}$. Figure 4 shows the annual concentrations and MFB of SO$_4$ at each station. Most stations give satisfactory results, 41 stations have a MFB between $\pm$

30% and 33 stations respect the goal criteria. The model gives no stations where SO$_4$ concentrations would be significantly underestimated (MFB<-30%) but results at some stations are significantly overestimated, especially in the North of Europe. This overestimation is similar to the overestimation of sea salts in Northern Europe. However, the contribution of sulfate from sea salts in the model (7.68 %) is not enough to explain the overestimation of sulfate in the Northern Europe. However, it may be due to an overestimation of north boundary conditions (as the stations in Northern Europe are close to the limit of the

domain), a lack of deposition or errors on meteorological data that create the same overestimation than for sea salts.

Figure 5 shows the seasonal evolution of the statistics for SO$_4$ by regions.

Like Na, SO$_4$ concentrations seem to be underestimated for the stations in the Southern Europe (mostly stations in Spain) in summer and overestimated in winter and late autumn with a MFB between -30% and 40%. However, this behavior is probably not due to sulfate from sea salts (due to the low contribution of sea salt to sulfate, only a small part of sulfate would be originat-





ing from sea salts). In Western, Central and Eastern Europe the model succeeds in reproducing the seasonal evolution with a MFB generally between ± 30% and MFE below 50% except for Western Europe in late autumn where MFB exceeds 50% and for Central Europe in November where MFB reaches 40%. $SO_4$ concentrations seem to be slightly relatively overestimated with a MFB>0 in Eastern Europe whereas MFB is between ± 30% for Central and Western Europe. In Northern Europe, like

sea salts, concentrations of $SO_4$ are overestimated with a MFB higher than 30% and reaching 90%.

### 3.3 Ammonium and nitrate

Annual scores for $NO_3$ and $NH_4$ are given in Table 5. Comparisons of $NO_3$ and $NH_4$ are carried out over 37 and 33 stations respectively. The model gives higher mean values than measurements (1.99 $\mu$g m$^{-3}$ against 1.46 $\mu$g m$^{-3}$ for $NO_3$ and 1.29

$\mu$g m$^{-3}$ against 0.89 $\mu$g m$^{-3}$ for $NH_4$. This kind of overestimation have been reported for numerous models (Bessagnet et al., 2014; Lecœur and Seigneur, 2013). RMSE are higher than mean measured concentrations (1.87 $\mu$g m$^{-3}$ and 0.97 $\mu$g m$^{-3}$) due to the high bias. The performance criteria are respected but not the goal criteria for both $NO_3$ (MFB = 15% and MFE = 57%) and $NH_4$ (MFB = 36% and MFE = 55%). However, the spatiotemporal correlation is rather high (0.71 for $NO_3$ and 0.71 for $NH_4$). Figure 6 shows the annual concentrations and MFB of $NO_3$ and $NH_4$ at each station. Both, $NO_3$ and $NH_4$ are

overestimated at some stations in Germany, one station near Barcelona and two stations in Eastern Europe indicating there may be too much ammonium nitrate at these stations. $NH_4$ is strongly overestimated in Northern Europe which may be linked to the overestimation of sulfate and the formation of ammonium sulfate over this region whereas $NO_3$ is overestimated at some stations in Southern Europe.

Figures 7 and 8 show the seasonal evolution of the statistics for $NO_3$ and $NH_4$ by regions.

For Southern Europe, both $NO_3$ and $NH_4$ concentrations are overestimated significantly in November-December 2013 with MFB exceeding 40% for $NO_3$ and 60% for $NH_4$. Concentrations of $NH_4$ are also overestimated from January to May with a MFB higher than 40% whereas $NO_3$ is overestimated to a smaller extent. $NO_3$ is also a bit underestimated from June to August. These results may indicate the formation of too high ammonium nitrate concentrations at the end of the year. For $NH_4$, errors on concentrations seem related to the errors on $SO_4$ concentrations indicating that $NH_4$ may be better represented with a

better representation of sulfates. Monthly correlations are low for $NH_4$ (lower than 0.4) and slightly higher for $NO_3$ (between 0.4 and 0.6).

For Western Europe, results on $NO_3$ are very similar to the results for $NH_4$ with an overestimation of both $NO_3$ and $NH_4$ in November and December (MFB higher than 40% for $NO_3$ and higher than 60% for $NH_4$) and a slight overestimation for the peak in March (MFB around 30% for both $NO_3$ and $NH_4$). All together the MFB for $NH_4$ is higher than the MFB for

$NO_3$ which may be due to the slight overestimation of $SO_4$ (and therefore the overestimation of ammonium sulfate). Monthly correlations are a bit higher for $NO_3$ (higher than 0.8 for most of the year) than for $NH_4$ (between 0.6 and 0.8 for most of the year).

Over Central Europe, $NO_3$ and $NH_4$ concentrations are strongly overestimated at the end of the year where high concentrations are simulated. $NH_4$ is also slightly overestimated at the beginning of the year with a MFB higher than 30%. Monthly




correlations are high (between 0.6 and 0.8) for $NO_3$ whereas monthly correlations are lower for $NH_4$ in summer (below 0.5).

The results over Eastern Europe are similar to the results of Central Europe, however $NO_3$ concentrations are overestimated at the beginning of the year and underestimated in Summer.

For Northern Europe, $NH_4$ concentrations are overestimated throughout the year with a MFB around 40% in May to August and up to 120% at the end of the year. These results are very similar to the results for $SO_4$. $NO_3$ concentrations are underestimated in summer and are strongly overestimated for the rest of the year (especially in February with a MFB close to 100%).

Generally, statistics for $SO_4$ and $NO_3$ seem better than the statistics for $NH_4$, this may be due to the cumulative errors on ammonium nitrate and ammonium sulfate. To investigate the highlighted results on ammonium and nitrate, scores were computed for total nitrate $TNO_3$ (particulate $NO_3$ + gaseous $HNO_3$) and for total ammonium $TNH_4$ (particulate $NH_4$ + gaseous $NH_3$). The performances are close to the goal criteria with a slight overestimation of concentrations. The mean seasonal evolutions of $TNO_3$ and $TNH_4$ are plotted in Figure 9. $TNO_3$ and $TNH_4$ share the same pattern with a slight underestimation of concentrations in summer and an overestimation of concentrations in autumn and winter.

This feature could be explained by:

– An overestimation of the gas-particle conversion of $HNO_3$ and $NH_3$. Indeed, Peters and Bruckner-Schatt (1995) measured higher deposition velocity of $HNO_3$ and $NH_3$ over spruce stand and Seinfeld and Pandis (1998) reported higher deposition velocity for gases than for particles lower than 2.5 $\mu$m over water surfaces. An overestimation of the partitioning can therefore lead to an overestimation of total concentrations because the deposition velocity of these gases are generally higher than those of particles (Peters and Bruckner-Schatt, 1995).

– An underestimation of the deposition velocity.

– An overestimation of $HNO_3$ production rate by the gas-phase mechanism MELCHIOR 2, leading to an overestimation of the partitioning of $NH_3$ and $HNO_3$ toward the particle phase and therefore leading to an overestimation of both $TNO_3$ and $TNH_4$.

– An overestimation of $NH_3$ emissions in winter and autumn and an underestimation of $NH_3$ emissions in summer. An overestimation of $NH_3$ would lead to an overestimation of the partitioning of $NH_3$ and $HNO_3$ toward the particle phase and therefore lead to an overestimation of both $TNO_3$ and $TNH_4$.

The last assumption seems to be supported by the shape of the seasonal profile of $NH_3$ emissions used in CHIMERE illustrated in Figure 10 which gives high emissions in November-December whereas Skjøth et al. (2011) (who developed a dynamical method to estimate $NH_3$ emissions based on the different types of agriculture) estimated very low emissions during this period. Using dynamical emissions may give a better representation of $NH_3$ and $NO_3$ concentrations. Moreover, in some countries from Northern Europe like Sweden, $NH_3$ emissions are mainly due to livestock (87 % of ammonia emissions in Sweden) whereas the temporal profile for Sweden is similar to the one of other countries with two peaks of emissions (one in March-April and one in October-November) corresponding to the application of fertilizers in Spring and Autumn. It could explain why for Northern



Europe, NO$_3$ is significantly underestimated in summer and significantly overestimated at the beginning and the end of the year.

### 3.3.1 Organic aerosol

Organic Aerosol concentration measurements are not available in the database. However, measurements for organic carbon (OC) concentrations are available. OC is the mass of carbon inside the organic aerosols. For the comparison, OM/OC ratios (that depend on the composition of organic aerosols, especially the degree of oxidation of compounds) have to be assumed to estimate OC concentrations from modeled OM concentrations. Turpin and Lim (2001) measured the OM/OC ratios at different locations and found ratios between 1.2 and 2.5 and recommended to use a ratio of 2.1 for rural areas. Following Couvidat et al. (2012), OC concentrations were calculated directly from the concentrations of each organic surrogate compounds using their molecular structure to estimate the OM/OC ratio of the surrogate compounds. Several sensitivity tests were conducted by Couvidat et al. (2012) and has shown that the OM/OC simulated by the H$^2$O mechanism is generally quite low compared to the OM/OC ratio recommended by Turpin and Lim (2001). An overestimation of OC concentrations by the model could therefore be due to an underestimation of the OM/OC ratio.

Table 6 shows the annual statistics for organic carbon (OC) for each station. Time series of the concentrations for each station are shown in Figures 12 and 13 for stations in the southern half and northern half of Europe respectively. Figure 11 shows the maps of OC concentrations over Europe for January and July 2013 as well as the MFB at stations. These figures show for January and July an underestimation of OC concentrations over Central Europe.

Annual concentrations at some stations seem to be overestimated (ES1778R, IT0004R and DE0003R) with MFB between 36% and 40%. However, the performance criteria is respected for these stations. It could also be possible that the overestimation of OC concentrations at these stations do not correspond to an underestimation of OC if the OM/OC ratio is underestimated. The performance goal is respected for stations CH0002R and SI0008R whereas the performance criteria are respected for stations CY0002R, DE0008R, PL0005R and SE0011R. Concentrations are underestimated at the other stations with MFB between -56% and -87%.

Although OC concentrations are slightly overestimated in ES1778 (near Barcelona, Spain) and IT0004 (in Ispra, Italy) the seasonality of OC concentrations is well captured by the model. Moreover, the overestimation of concentrations could be due to the proximity of high emissions sources and the low resolution of the model (ES1778 is only at 50 km from Barcelona whereas the resolution is only of 0.25° and IT0004 is close to Milan). Concentrations and the seasonal evolution are well reproduced in SI0008 (Iskbra, Slovenia). For the other stations in the south of Europe, summer concentrations are underestimated in summer whereas the winter concentrations seem to be well reproduced. This may indicate a lack of secondary organic aerosol formation. These stations all have strong modeled concentrations of biogenic SOA in summer and strong modeled concentrations of anthropogenic organic aerosol in winter.

For the northern half of Europe, except for stations DE0003 (for which concentrations are overestimated in Winter), the station CH0005R in Switzerland and PL0005R in Poland, summer concentrations of organic aerosol are underestimated. Only





a peak of organic aerosol (due to biogenic aerosols in the model) at the end of August for several stations (CZ0003R, DE0002R, DE0007R, DE0008R, DE0044R) is well reproduced by the model.

### 3.3.2 PM concentrations

Annual scores for $PM_{2.5}$ and $PM_{10}$ are given in Table 5. Comparisons are carried out over 41 stations for $PM_{2.5}$ and 59 stations for $PM_{10}$. The goal criteria is respected for both $PM_{2.5}$ and $PM_{10}$. However, $PM_{2.5}$ concentrations are slightly overestimated (MFB = 22%). The MFB for $PM_{10}$ is lower (8%) indicating that coarse particles may be underestimated. This is confirmed by the fact that for the stations with measurements of both $PM_{2.5}$ and $PM_{10}$, the MFB for $PM_{2.5}$ (17%) is higher than the MFB for $PM_{10}$ (4%). The underestimation of coarse particles was reported for numerous models in the intercomparison model projet AQMEII (Solazzo et al., 2012; Pirovano et al., 2012).

The simulated mean concentrations (10.54 $\mu$g m$^{-3}$ for $PM_{2.5}$ and 14.42 $\mu$g m$^{-3}$ for $PM_{10}$) are close to the measured mean concentrations (9.06 $\mu$g m$^{-3}$ for $PM_{2.5}$ and 13.51 $\mu$g m$^{-3}$ for $PM_{10}$) and the spatiotemporal correlations are high (0.68 for $PM_{2.5}$ and 0.60 for $PM_{10}$).

Figure 14 shows the annual concentrations and MFB of $PM_{2.5}$ and $PM_{10}$ at each station. The model strongly underestimates annual concentrations of $PM_{10}$ only for the ES0008R station, probably due to the underestimation of sea salt at this station. However, the model overestimates PM concentrations with a MFB above 30% at several locations (14 stations for $PM_{2.5}$ and 12 stations for $PM_{10}$ especially over the Alps).

Figures 15 and 17 show the seasonal evolution of the statistics by regions for $PM_{2.5}$ and $PM_{10}$, respectively.

$PM_{2.5}$ concentrations seem to be underestimated for the stations in the Southern Europe from June to August and overestimated the rest of the year (especially in March, November and December with MFB reaching 60%). A similar feature is obtained with $PM_{10}$ but with lower MFB in March, November and December, which is probably due to some compensation effects and an underestimation of the coarse fraction of PM. Based on these results, this overestimation is probably mainly due to the overestimation of ammonium nitrate observed during these months while the underestimation from June to August is probably due to the underestimation of all PM components.

For Western Europe, $PM_{2.5}$ are overestimated from September to December with a MFB between 40% and 60% which may be due at least partly to the overestimation of ammonium nitrate. The overestimation could also be due to an overestimation of organic matter observed at some stations or to an overestimation of primary particles. Similar results are obtained for $PM_{10}$.

In Central Europe, the model reproduces well the strong concentrations of $PM_{2.5}$ and $PM_{10}$ in Winter and Spring (with a MFE around 40%). However, concentrations are slightly underestimated in summer (with a negative MFB reaching -30% for $PM_{2.5}$ and -40% for $PM_{10}$) and are overestimated from October to December with a MFB between 40% and 80% for $PM_{2.5}$ (probably due to the strong overestimation of ammonium nitrate) and reaching 50% for $PM_{10}$.

For Eastern and Northern Europe, similar features are obtained. $PM_{2.5}$ and $PM_{10}$ are overestimated in Winter and Fall (probably due to mostly to the overestimation of ammonium nitrate) and underestimated in Summer (probably due to the summer underestimation of ammonium nitrate and organic aerosols).

A Quantile-Quantile (QQ) scatter plot of modeling results against measurements for $PM_{10}$ and $PM_{2.5}$ is shown in Figure





16. QQ plots can be used to assess the similarity of the distribution of two compared datasets. For Southern Europe, concentrations higher than 40 $\mu$g m$^{-3}$ for PM$_{10}$ and higher than 20 $\mu$g m$^{-3}$ for PM$_{2.5}$ are significantly underestimated, especially for high concentrations of PM$_{2.5}$. Western Europe and Central Europe have a similar distribution of concentrations with a strong overestimation of high concentrations of PM$_{2.5}$ (higher than 35 $\mu$g m$^{-3}$) and of PM$_{10}$ (higher than 55 $\mu$g m$^{-3}$). This overestimation is probably due to the high overestimation of ammonium nitrate during the late autumn for these regions. For Eastern Europe and Southern Europe, the distributions of modeled concentrations are similar to the distributions of observed concentrations.

### 3.3.3 Case of the Cyprus station

The Cyprus station (CY0002R) was analyzed due to the specificities of this station close to the boundary conditions, influenced by high concentrations of PM due to mineral dusts and high anthropogenic emissions from the Mediterranean maritime traffic. Morever, numerous measurements were carried out at this station: PM$_{2.5}$ and PM$_{10}$ and measurements of speciation (both in the fine fraction and in PM$_{10}$) covering NO$_3$, NH$_4$, SO$_4$, Na, Cl and also Ca (originating mainly from dust). The temporal evolution of PM$_{2.5}$ and PM$_{10}$, Ca (fine fraction and total), NO$_3$ (fine fraction and total) are shown in Figure 18 and the temporal evolution of Na, Cl, NH$_4$ and SO$_4$ are shown in Figure 19. The temporal evolution of OC concentrations is shown in Figure 13.

The model gives at this station good performances for the simulation of PM$_{2.5}$ (correlation = 0.54, MFB=-10%, MFE=35%) and PM$_{10}$ (correlation=0.64, MFB=26%, MFE=39%). The temporal evolution of PM$_{2.5}$ and PM$_{10}$ are well reproduced by the model. The good results at this station are mainly due to the good representation of dusts transport in the simulation (coming here from the boundary conditions taken from Mozart v4.0). Simulated concentrations of Ca (assuming a fraction of 6% in dusts) were compared to the measurements of Ca. The model gives good results for Ca concentrations in PM10 (correlation=0.65, MFB=6% and MFE=45%) but Ca concentrations in fine particles are overestimated (correlation=0.53, MFB=63% and MFE=81%) but the concentrations (0.19 $\mu$g m$^{-3}$ in measurements) are low compared to Ca in PM10 (1.32 $\mu$g m$^{-3}$ in measurements). Na and Cl are strongly underestimated (MFB=-85% and MFB=-40% for Cl).

The model almost respects the goal criteria for NO$_3$ in PM$_{10}$ (MFB=24% and MFE=51%) but with a low correlation (0.38). The modeled annual mean of NO$_3$ in PM$_{10}$ (1.66 $\mu$g m$^{-3}$) is close to the measured annual mean (1.53 $\mu$g m$^{-3}$). However, the measured annual mean of NO$_3$ in fine particles is strongly overestimated (0.71 $\mu$g m$^{-3}$ against 0.18 $\mu$g m$^{-3}$). The model seems here to underestimate the coarse faction of NO$_3$. Even if the model gives strong NO$_3$ concentrations in the coarse fraction due to the coarse mode formation with dusts, a significant part of NO$_3$ in the model seems to be due to ammonium nitrate formation in PM whereas most of NH$_4$ seems to be due to ammonium sulfate formation (correlation of 0.85 between observed sulfate and ammonium). A similar feature is observed at station ES0008R with a good order of magnitude for NO$_3$ in PM$_{10}$ (1.23 $\mu$g m$^{-3}$ in the model and 1.19 $\mu$g m$^{-3}$ in measurements) but with an overestimation of NO$_3$ in the fine fraction (1.22 $\mu$g m$^{-3}$ in the model and 0.38 $\mu$g m$^{-3}$ in measurements). These results may be partly due to a lack of HNO$_3$ condensing onto dusts but the underestimation is probably mainly due to sea salts which are underestimated at these two stations.

NH$_4$ concentrations are overestimated (probably due to the overestimation of ammonium nitrate and the lack of HNO$_3$ con-



densing onto dusts) with a MFB of 49% whereas $SO_4$ concentrations are underestimated (MFB=-0.29%) especially in July and November 2013 where the model is not able to reproduce the high concentrations of sulfate. Correlation are very low (0.23 for $NH_4$ and 0.13 for $SO_4$).

### 5  3.3.4  Case of the Melpitz station (Germany)

Numerous simultaneous types of measurements were also carried out at the Melpitz station (DE0044R) in Germany. The temporal evolution of $PM_{2.5}$ and $PM_{10}$, Ca (fine fraction and total), $NO_3$ (fine fraction and total) are shown in Figure 20 and the temporal evolution of Na, Cl, $NH_4$, $SO_4$ and elemental carbon (EC) are shown in Figure 21. The temporal evolution of OC concentrations is shown in Figure 12

At this station, $PM_{2.5}$ and $PM_{10}$ share a similar pattern. For $PM_{2.5}$, annual concentrations are underestimated by the model (13.2 $\mu$g m$^{-3}$ against 17.8 $\mu$g m$^{-3}$ in measurements) especially in summer with an underestimation ranging from 7 and 11 $\mu$g m$^{-3}$ from June to August and a monthly MFB between -65% and -95% and in a lesser extent in winter (except in February) with an underestimation of about 7 $\mu$g m$^{-3}$. For $PM_{10}$, the underestimation is stronger (15.7 $\mu$g m$^{-3}$ against 22.1 $\mu$g m$^{-3}$) especially between april and august with an underestimation between 10 and 15 $\mu$g m$^{-3}$.

Concentrations of $SO_4$ are well reproduced by the model with a high correlation (0.83) and low MFB and MFE (0.12% and 0.35%). The model succeeds to capture the high concentrations of $SO_4$ in winter. $NO_3$ concentrations and $NH_4$ concentrations are also well reproduced by the model except at the end of the year where concentrations are strongly overestimated. The high concentrations of $NO_3$ in February and March are well reproduced by the model (temporal correlation of 0.91 and 0.73, MFE=30% and 40%). The low concentrations of Na and Cl are a bit overestimated. The overestimation reaches 0.34 $\mu$g m$^{-3}$

for Na and 0.41 $\mu$g m$^{-3}$ for Cl in January.

Most of the underestimation of $PM_{2.5}$ concentrations is probably due to the underestimation of organic aerosols at this station. Indeed, OC concentrations are underestimated by 2.9 $\mu$g m$^{-3}$ from June to August. Using the OM/OC ratio of 2.1 measured by Turpin and Lim (2001) for rural areas, the underestimation of organic aerosol could explain most of the differences between modeled and measured $PM_{2.5}$. However, in June, the underestimation of OC concentrations is only of 3.3 $\mu$g

m$^{-3}$ which would correspond to an underestimation of OM of 7 $\mu$g m$^{-3}$ whereas $PM_{2.5}$ concentrations are underestimated by 11 $\mu$g m$^{-3}$. It appears difficult that in June, the underestimation alone of organic aerosols could explain all the underestimation of PM. The remaining underestimation in June cannot be explained by $SO_4$, $NO_3$, $NH_4$, Na or Cl. The sum of these concentrations is 4.0 $\mu$g m$^{-3}$ for the model and 3.0 $\mu$g m$^{-3}$ for measurements. The fine fraction of Ca is a bit underestimated by the model (0.08 $\mu$g m$^{-3}$ against 0.11 $\mu$g m$^{-3}$ in measurements), which could explain an underestimation of 0.75 $\mu$g m$^{-3}$

(assuming that there is 4% of Ca in dusts) which is just enough to compensate the overestimation by the model of inorganic aerosols. The remaining underestimation in June could be explain by an underestimation of primary aerosols. However, EC concentrations are well reproduced by the model (annual correlation of 0.62, MFB=0.34% and MFE=47%) and are slightly overestimated by the model (0.42 $\mu$g m$^{-3}$ against 0.22 $\mu$g m$^{-3}$ in measurements).



## 4   Perspectives on model improvement

The following list provides a list of possible developments that may be addressed in the future:

- The formation of ammonium nitrate in the model for which the strongest errors were obtained need to be improved. As NH$_3$ emissions is a key element, implementing a dynamic method to improve the spatial and temporal evolution of NH$_3$ emissions from agriculture like the method described in Skjøth et al. (2011) depending on temperature and wind speed could be a major improvement.

- The formation of anthropogenic SOA has to be better represented in the model. For that, SVOC and IVOC emissions should be better represented. The inventory of Denier van der Gon et al. (2015) could be used to better estimate SVOC from residential biomass burning. More generally, to take into IVOC emissions, emission inventory by volatility classes should be developed based for example on the method developed by Zhao et al. (2016b). Moreover, mechanisms of formation of SOA from IVOC oxidation and the aging of SVOC have to be better understood. Bruns et al. (2016) showed that the formation of SOA from SOA precursors traditionally taken into account in models (like toluene, xylene, alkanes) only amount for a small amount of SOA (between 3 and 27% of SOA formed) from biomass burning and that most of the SOA is due to non-traditional SOA precursors (like phenol, naphthalene, benzaldehyde, etc...). These precursors should be added in the SOA mechanism. Aging has also a major impact on SOA formation (Donahue et al., 2012; Zhao et al., 2016a) and should be studied into greater details.

- The influence of the gas-phase mechanism on PM formation should be tested within CHIMERE. Indeed, Sarwar et al. (2013) found significant differences over the United States of America between CB05 (Sarwar et al., 2008) and RACM2 (Goliff et al., 2013) in OH concentrations (46% in OH concentrations) and PM components (10 % in sulfate, 6% in nitrate, 10% in ammonium, 42% in anthropogenic SOA and 5% in biogenic SOA). The strong differences in radical concentrations may strongly affect the aging of SVOC compounds and the seasonal evolution of PM components. It may also be important to compare the results of the MELCHIOR 2 mechanisms with more recent gas-phase mechanisms.

- The formation of inorganic aerosol could be integrated in the SOAP thermodynamic model. SOAP would then be able to simulate both inorganic and organic aerosols and take into account the influence between inorganic and organic aerosols which can affect the partitioning of compounds and the hygroscopicity of the aerosol (Jing et al., 2016). This could also be important to take into account the formation of some acid organics/ammonium salts which can be in competition with ammonium nitrate formation. Some organonitrogen compounds were also found by condensation of ammonia onto organic aerosols (Liu et al., 2015).

- Some recent experimental studies emphasize the need to account for dynamical aspects of the organic aerosols formation rather than assuming thermodynamic equilibrium with the gas phase because organic aerosols can be highly viscous (Virtanen et al., 2010; Cappa and Wilson, 2011; Pfrang et al., 2011; Shiraiwa et al., 2011; Vaden et al., 2011; Shiraiwa and Seinfeld, 2012; Abramson et al., 2013). To our knowledge, this phenomenon was never investigated inside a 3D air



quality model. However, to take it into account, a dynamic method for SOA formation taking into account the diffusion inside a few layers was developed in the thermodynamic model SOAP. This method could be used to test the influence of the organic-phase viscosity on SOA formation inside a 3D air quality model.

– Aqueous-phase chemical mechanism can be extensively improved. Several studies highlight the importance of aqueous-phase chemistry for SOA formation from isoprene. An isoprene-derived epoxidiol (IEPOX) has been shown to form methyltetrols and $C_5$-alkene triols in the aqueous phase of particles and droplets by hydrolysis as well as organosulfates by reaction with sulfate or bisulfate ions and oligomers (Surratt et al., 2010). Froyd et al. (2010) found very high concentrations of SOA formed from IEPOX in Atlanta, USA (910 ng.m$^{-3}$), due to a very acidic aerosol. The formation of SOA from IEPOX was investigated in a previous study (Couvidat et al., 2013). Using a Henry's law constant of IEPOX of $2 \times 10^7$ M/atm and a mechanism based on available information, the model could simulate concentrations of SOA from IEPOX in the right order of magnitude and simulates concentrations of SOA from IEPOX in summer that could could reach 1 $\mu$g m$^{-3}$ over some regions and could give strong peaks of SOA. Nguyen et al. (2014) found a Henry's law constant of $3 \times 10^7$ M/atm. However, this modeling study did not take into account interactions with inorganics aerosol. Aqueous-phase processing of glyoxal (which is formed from the oxidation of toluene and isoprene) was also found to be possibly a significant source of SOA as it can form oxalic acid via reaction in clouds (Griffin et al., 2003) and could form oligomers or form SOA in particles via reaction with hydroxyl radical or from reactions catalysed by ammonium (Knote et al., 2014) which could be important to take into account. Implementing a complete cloud chemical mechanism should be tested. A mechanism similar to Leriche et al. (2013) could be implemented to improve the representation of the cloud chemistry inside the model.

– The number of particles has also an effect on health, especially in urban areas (Jing et al., 2001). To properly estimate the effects on health of particles in urban areas, more complete parameterization of nucleation should be implemented (taking for example the impact of organic compounds (Lupascu et al., 2015)) and model results should be compared to available data on European Megacities.

– The possibility to take into account the mixing state (as done by Zhu et al. (2015, 2016); S. et al. (2016) should be added to the model as it can impact the aerosol formation and composition and their optical and hygroscopic properties.

## 5 Conclusions

Concentrations were compared to available information on PM concentrations and composition thanks to the EBAS database. Whereas the model gives satisfactory results in regard of the criteria defined by Boylan and Russell (2006), results could be improved in terms of seasonality and PM composition. Strongest errors were found to be probably due to ammonium nitrate which is often overestimated especially in late autumn (probably due to an overestimation of $NH_3$ emissions during this period). Only in summer, concentrations of ammonium nitrate could be underestimated. Strong errors were also found on OC concentrations in summer (especially over the northern half of Europe) indicating that strong concentrations of anthropogenic





SOA could be missing from the models. Sea salts concentrations were properly simulated at most stations but were overestimated over regions with low concentrations and underestimated for regions with very strong concentrations (which could lead to an underestimation of $HNO_3$ condensation onto coarse particles).

However, the model has good performances in general and respects the goal criteria for both $PM_{2.5}$ and $PM_{10}$. For sea salts, the model almost respects the goal criteria of Boylan and Russell (2006) and succeed in reproducing the seasonal evolution of concentrations for Western and Central Europe. For sulfate, except for an overestimation of sulfate in Northern Europe, modeled concentrations are close to observations with a good seasonal evolution of concentrations. For organic aerosol, the model performs well for stations with strong modeled biogenic SOA concentrations.

Several improvements should be tested. Taking into account the dynamic of $NH_3$ emissions could greatly improve the results of the model in simulating ammonium nitrate. Taking into account SOA formation from missing SVOC/IVOC emissions probably has an important impact on SOA formation and could probably improve results on SOA concentrations. Moreover, the impact of inorganic-organic interactions and the effect of aqueous-phase chemistry on SOA formation should be investigated.

## 6  Code availability

Chimere2017$\beta$ is a version of chimere developed for research on aerosol modeling. It is available on request by contacting the authors.

*Acknowledgements.*  This work was funded by the French Ministry of the Environment, Energy and Marine Affairs. Simulations were performed using the TGCC-CCRT super computers under the GENCI time allocation gen7485.



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




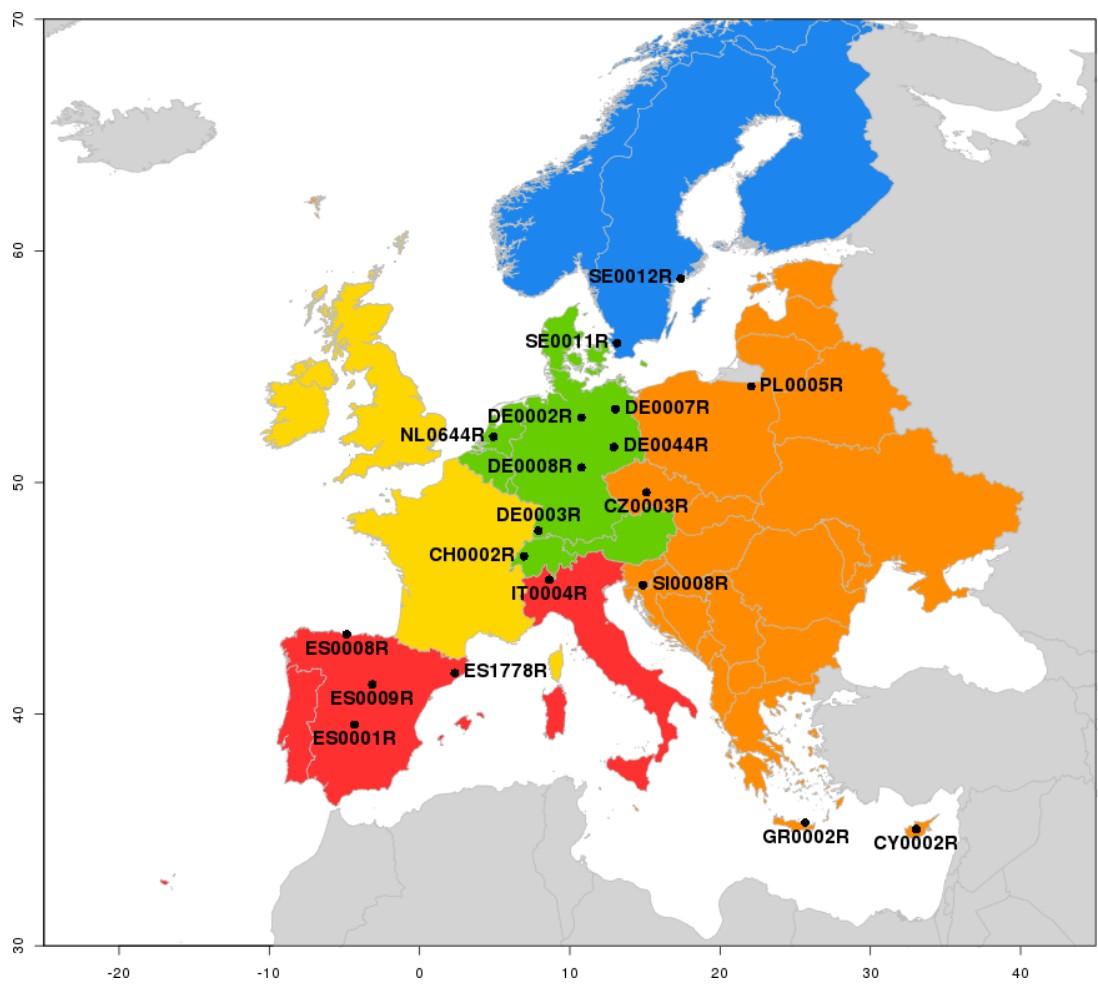

**Figure 1.** Maps of regions defined in this study and of mentionned stations. The chosen regions are: Southern Europe (Red), Western Europe (Yellow), Central Europe (Green), Eastern Europe (Orange), Northern Europe (Blue).





**Table 1.** Reactions leading to SOA formation[a].

| Reaction | Kinetic rate parameter |
|---|---|
| | $(s^{-1} \text{ or molecule}^{-1}.cm^3.s^{-1})$ |
| ISOP + OH → ISOR + OH | $2.54 \times 10^{-11} \times \exp(\frac{408}{T})$ |
| ISOP + NO$_3$ → ISON + NO$_3$ | $3.03 \times 10^{-12} \times \exp(\frac{-448}{T})$ |
| ISOR + HO$_2$ → 0.282 BiPER + 0.030 BiDER + HO$_2$ | $2.05 \times 10^{-13} \times \exp(\frac{1300}{T})$ |
| ISOR + C$_2$O$_3$ → 0.026 BiMT + 0.219 MACR + C$_2$O$_3$ | $8.40 \times 10^{-14} \times \exp(\frac{221}{T})$ |
| ISOR + MeO$_2$ → 0.026 BiMT + 0.219 MACR + MeO$_2$ | $3.40 \times 10^{-14} \times \exp(\frac{221}{T})$ |
| ISOR + NO → 0.418 MACR + 0.046 ISON + NO | $2.43 \times 10^{-12} \times \exp(\frac{360}{T})$ |
| ISOR + NO$_3$ → 0.438 MACR + NO$_3$ | $1.20 \times 10^{-12}$ |
| ISON + OH → OH | $1.30 \times 10^{-11}$ |
| ISON + NO$_3$ → 0.074 BiNIT3 + NO$_3$ | $6.61 \times 10^{-13}$ |
| MACR + NO → NO | $2.54 \times 10^{-12} \times \exp(\frac{360}{T})$ |
| MACR + HO$_2$ → HO$_2$ | $1.82 \times 10^{-13} \times \exp(\frac{1300}{T})$ |
| MACR + MeO$_2$ → MeO$_2$ | $3.40 \times 10^{-14} \times \exp(\frac{221}{T})$ |
| MACR + NO$_2$ → MPAN + NO$_2$ | $2.80 \times 10^{-12} \times \exp(\frac{181}{T})$ |
| MPAN → MACR | $1.60 \times 10^{16} \times \exp(\frac{-13486}{T})$ |
| MPAN + OH → 0.067 BiMGA + 0.047 BiNGA + OH | $3.20 \times 10^{-11}$ |
| MPAN + NO$_3$ → 0.067 BiMGA + 0.047 BiNGA + NO$_3$ | $3.20 \times 10^{-11}$ |
| BiPER + h$\nu$ → Degradation products | k = 50 × kinetic of photolysis of H$_2$O$_2$ |





| Reaction | Kinetic rate parameter $(s^{-1}$ or molecule$^{-1}$.cm$^3$.s$^{-1})$ |
|---|---|
| API + OH → 0.30 BiA0D + 0.17 BiA1D + 0.10 BiA2D + OH | $1.21 \times 10^{-11} \times \exp(\frac{440}{T})$ |
| API + O$_3$ → 0.18 BiA0D + 0.16 BiA1D + 0.05 BiA2D + O$_3$ | $5.00 \times 10^{-16} \times \exp(\frac{-530}{T})$ |
| API + NO$_3$ → 0.70 BiA0D + 0.10 BiNIT + NO$_3$ | $1.19 \times 10^{-12} \times \exp(\frac{-490}{T})$ |
| BPI + OH → 0.07 BiA0D + 0.08 BiA1D + 0.06 BiA2D + OH | $2.38 \times 10^{-11} \times \exp(\frac{357}{T})$ |
| BPI + O$_3$ → 0.09 BiA0D + 0.13 BiA1D + 0.04 BiA2D + O$_3$ | $1.50 \times 10^{-17}$ |
| BPI + NO$_3$ → 0.02 BiA0D + 0.63 BiNIT + NO$_3$ | $2.51 \times 10^{-12}$ |
| LIM + OH → 0.35 BiA0D + 0.20 BiA1D + 0.0035 BiA2D + OH | $4.20 \times 10^{-11} \times \exp(\frac{401}{T})$ |
| LIM + O$_3$ → 0.09 BiA0D + 0.10 BiA1D + O$_3$ | $2.95 \times 10^{-15} \times \exp(\frac{783}{T})$ |
| LIM + NO$_3$ → 0.69 BiA0D + 0.27 BiNIT + NO$_3$ | $1.22 \times 10^{-11}$ |
| HUM + OH → 0.74 BiBmP + 0.26 BiBlP + OH | $2.93 \times 10^{-10}$ |
| TOL + OH → ... + 0.25 TOLP | $1.80 \times 10^{-12} \times \exp(\frac{355}{T})$ |
| TOLP + HO$_2$ → 0.78 AnClP + HO$_2$ | $3.75 \times 10^{-13} \times \exp(\frac{980}{T})$ |
| TOLP + C$_2$O$_3$ → 0.78 AnClP + C$_2$O$_3$ | $7.40 \times 10^{-13} \times \exp(\frac{765}{T})$ |
| TOLP + MeO$_2$ → 0.78 AnClP + MeO$_2$ | $3.56 \times 10^{-14} \times \exp(\frac{708}{T})$ |
| TOLP + NO → 0.097 AnBlP + 0.748 AnBmP + NO | $2.70 \times 10^{-12} \times \exp(\frac{360}{T})$ |
| TOLP + NO$_3$ → 0.097 AnBlP + 0.748 AnBmP + NO$_3$ | $1.2 \times 10^{-12}$ |
| XYL + OH → ... + 0.274 XYLP | $1.70 \times 10^{-11} \times \exp(\frac{116}{T})$ |
| XYLP + HO$_2$ → 0.71 AnClP + HO$_2$ | $3.75 \times 10^{-13} \times \exp(\frac{980}{T})$ |
| XYLP + C$_2$O$_3$ → 0.71 AnClP + C$_2$O$_3$ | $7.40 \times 10^{-13} \times \exp(\frac{765}{T})$ |
| XYLP + MeO$_2$ → 0.71 AnClP + MeO$_2$ | $3.56 \times 10^{-14} \times \exp(\frac{708}{T})$ |
| XYLP + NO → 0.063 AnBlP + 0.424 AnBmP + NO | $2.70 \times 10^{-12} \times \exp(\frac{360}{T})$ |
| XYLP + NO$_3$ → 0.063 AnBlP + 0.424 AnBmP + NO$_3$ | $1.2 \times 10^{-12}$ |

[a] Oxidants may be present as both reactants and products so that a reaction added to CB05 will not affect the original photochemical oxidant concentrations.




**Table 2.** Properties of the surrogate SOA species.

| Surrogate | Type [a] | H [b] | $P^{0}$ [c] | $\Delta H_{vap}$ [d] | Comments |
|---|---|---|---|---|---|
| BiMT | hydrophilic | 0.805 | $1.45 \times 10^{-6}$ | 38.4 | - |
| BiPER | hydrophilic | 0.111 | $2.61 \times 10^{-6}$ | 38.4 | - |
| BiDER | hydrohilic | 2.80 | $4.10 \times 10^{-7}$ | 38.4 | - |
| BiMGA | hydrophilic | $1.13 \times 10^{-2}$ | $1.4 \times 10^{-5}$ | 43.2 | $pK_a = 4.0$ |
| BiNGA | hydrophobic | - | $1.4 \times 10^{-5}$ | 43.2 | $K_{p,eff}=K_p(1+K_{oligo})$ [e] |
| BiNIT3 | hydrophobic | - | $1.45 \times 10^{-6}$ | 38.4 | - |
| BiA0D | hydrophilic | $4.82 \times 10^{-5}$ | $2.70 \times 10^{-4}$ | 50 | See Eq. 4 |
| BiA1D | hydrophilic | $2.73 \times 10^{-3}$ | $2.17 \times 10^{-7}$ | 50 | $pK_a = 3.2$ |
| BiA2D | hydrophilic | $6.52 \times 10^{-3}$ | $1.43 \times 10^{-7}$ | 50 | $pK_{a1} = 3.4, pK_{a2} = 5.1$ |
| BiNIT | hydrophobic | - | $2.5 \times 10^{-6}$ | 109 | - |
| BiBlP | hydrophobic | - | $6.0 \times 10^{-10}$ | 175 | - |
| BiBmP | hydrophobic | - | $3.0 \times 10^{-7}$ | 175 | - |
| AnBlP | hydrophobic | - | $6.8 \times 10^{-8}$ | 50 | - |
| AnBmP | hydrophobic | - | $8.4 \times 10^{-6}$ | 50 | - |
| AnClP | hydrophobic | - | - | non volatile | - |

[a] Type A: hydrophilic species, type B: hydrophobic species, type C: hydrophobic non-volatile species, which is not used to compute activity coefficients

[b] Henry's law constant [$(\mu g/\mu g$ water$)/(\mu g/m^3)$]

[c] Saturation vapor pressure [torr]

[d] Enthalpy of vaporization [kJ.mol$^{-1}$]

[e] $K_{oligo}$ (equal to 64.2) is used to take into account the formation of oligomers (Couvidat et al., 2012). $K_{p,eff}$ is the effective partitioning constant and $K_p$ is the partitioning constant calculated as in Pankow (1994).



**Table 3.** Properties of primary and aged SVOC.

| Surrogate | MW[a] | $K_p$ [b] | $\Delta H_{vap}$ [c] |
|-----------|-------|-----------|----------------------|
| POAlP | 280 | 1.1 | 106 |
| POAmP | 280 | 0.0116 | 91 |
| POAhP | 280 | 0.00031 | 79 |
| SOAlP | 392 | 110 | 106 |
| SOAmP | 392 | 1.16 | 91 |
| SOAhP | 392 | 0.031 | 79 |

[a] Molecular weight [g.mol$^{-1}$]

[b] Partitioning constant [m$^3$.$\mu$g$^{-1}$]

[c] Enthalpy of vaporization [kJ.mol$^{-1}$]

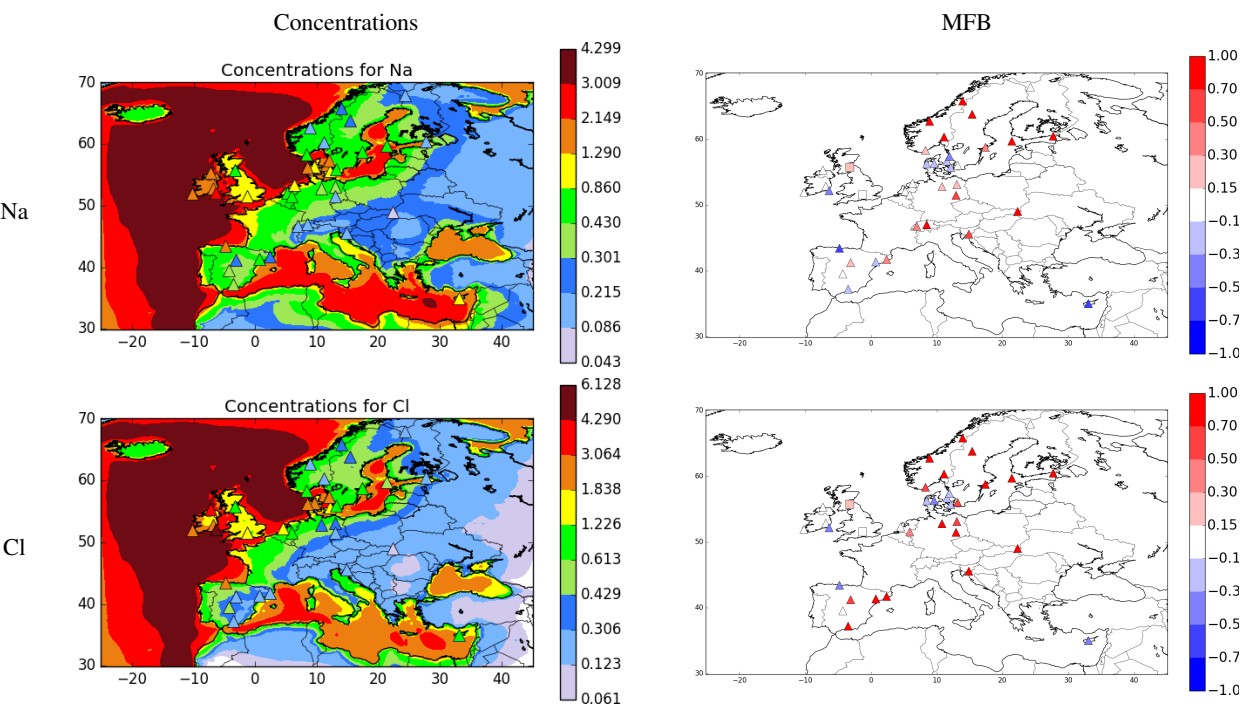

**Figure 2.** Modeled concentrations (in $\mu$g m$^{-3}$) and MFB for Na and Cl in 2013. Triangles correspond to measured concentrations in the left and to the MFB value in the right.





**Table 4.** Effective Henry's law constant used for dry and wet deposition.

| Compound | H (mol/L/atm) |
|----------|---------------|
| $O_3$ | 0.01 |
| $SO_2$ | $10^5$ |
| $NO_2$ | 0.01 |
| NO | $2 \times 10^{-3}$ |
| $NH_3$ | $10^5$ |
| BiA0D | 1.8e6 |
| BiA1D | 2.92e10 |
| BiA2D | 4.23e10 |
| BiMT | 3.3e10 |
| BiPER | 8.09e9 |
| BiDER | 8.91e10 |
| BiMGA | 2.15e10 |
| AnBlP | 0.01 |
| AnBmP | 0.01 |
| BiBlP | 1.17e8 |
| BiBmP | 3.05e5 |
| AnClP | 0.01 |
| BiNGA | 2.71e9 |
| BiNIT3 | 4.75e6 |
| BiNIT | 7.66e4 |
| POAlP | 0.01 |
| POAmP | 0.01 |
| POAhP | 0.01 |
| SOAlP | 3000 |
| SOAmP | 3000 |
| SOAhP | 3000 |





**Table 5.** Annual statistics for the comparison of daily concentrations. Means and RMSE are in $\mu g\ m^{-3}$.

|  | O3 | NO$_2$ | PM$_{10}$ | PM$_{2.5}$ | NO$_3$ | NH$_4$ | SO$_4$ | Na | Cl | TNO$_3$ | TNH$_4$ |
|---|---|---|---|---|---|---|---|---|---|---|---|
| Number of stations | 117 | 81 | 59 | 41 | 37 | 33 | 56 | 38 | 35 | 42 | 42 |
| Model mean | 71.78 | 6.83 | 14.42 | 10.54 | 1.99 | 1.29 | 1.66 | 0.67 | 1.28 | 2.55 | 2.16 |
| Measurement mean | 63.90 | 6.57 | 13.51 | 9.06 | 1.46 | 0.89 | 1.6 | 0.69 | 1.17 | 2.07 | 1.49 |
| RMSE | 19.62 | 6.4 | 9.34 | 6.95 | 1.87 | 0.97 | 1.13 | 0.76 | 1.49 | 2.0 | 2.0 |
| Correlation | 0.62 | 0.6 | 0.6 | 0.68 | 0.71 | 0.71 | 0.67 | 0.66 | 0.67 | 0.69 | 0.56 |
| MFB | 0.02 | -0.03 | 0.08 | 0.22 | 0.15 | 0.36 | 0.13 | 0.06 | 0.09 | 0.10 | 0.20 |
| MFE | 0.16 | 0.51 | 0.44 | 0.48 | 0.57 | 0.55 | 0.44 | 0.52 | 0.49 | 0.50 | 0.55 |

**Table 6.** Comparison of modeled concentrations to measured concentrations of OC. Means and RMSE are in $\mu g\ m^{-3}$.

| Station | Modeled mean | Measured mean | RMSE | correlation | MFB | MFE |
|---|---|---|---|---|---|---|
| CH0002R | 2.48 | 3.29 | 1.70 | 0.67 | -0.13 | 0.37 |
| CY0002R | 0.91 | 1.65 | 1.65 | 0.07 | -0.35 | 0.65 |
| CZ0003R | 1.97 | 3.64 | 2.28 | 0.72 | -0.63 | 0.67 |
| DE0002R | 1.20 | 2.40 | 1.79 | 0.40 | -0.68 | 0.75 |
| DE0003R | 1.83 | 1.27 | 1.28 | 0.35 | 0.40 | 0.63 |
| DE0007R | 1.28 | 2.26 | 2.03 | 0.29 | -0.56 | 0.73 |
| DE0008R | 1.49 | 1.73 | 1.18 | 0.49 | -0.03 | 0.54 |
| DE0044R | 1.67 | 3.65 | 2.45 | 0.82 | -0.74 | 0.76 |
| ES0001R | 0.90 | 1.79 | 1.18 | 0.64 | -0.65 | 0.70 |
| ES0009R | 0.69 | 1.79 | 1.87 | 0.07 | -0.77 | 0.87 |
| ES1778R | 1.97 | 1.60 | 1.07 | 0.68 | 0.36 | 0.48 |
| IT0004R | 10.4 | 6.36 | 7.43 | 0.84 | 0.40 | 0.55 |
| NL0644R | 1.21 | 2.57 | 1.53 | 0.84 | -0.87 | 0.87 |
| PL0005R | 1.87 | 3.04 | 2.18 | 0.71 | -0.42 | 0.49 |
| SI0008R | 4.70 | 4.19 | 2.45 | 0.53 | -0.01 | 0.40 |
| GR0002R | 0.82 | 1.78 | 1.42 | 0.19 | -0.65 | 0.72 |
| SE0011R | 0.84 | 1.14 | 0.71 | 0.41 | -0.32 | 0.50 |
| SE0012R | 0.73 | 1.53 | 1.16 | 0.74 | -0.70 | 0.72 |





**Figure 3.** Seasonal evolution of statistics by regions for Na: Monthly mean measured concentrations (black), monthly mean modeled concentrations (red), monthly RMSE (blue), monthly spatiotemporal correlations (green), monyhly MFB (cyan) and monthly MFE (magenta). Solid curves refer to the left axis while dotted curves refer to the right axis.





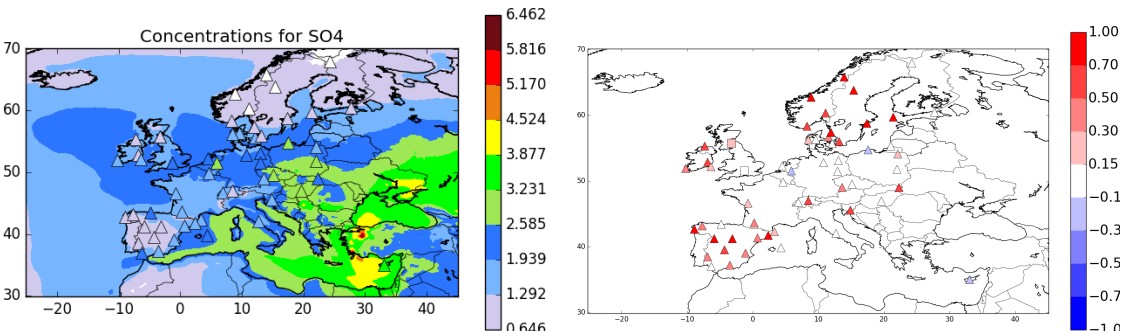

**Figure 4.** Modeled concentrations (in $\mu g\ m^{-3}$) and MFB for $SO_4$ in 2013. Triangles correspond to measured concentrations in the left and to the MFB value in the right.





**Figure 5.** Seasonal evolution of statistics by regions for $SO_4$: Monthly mean measured concentrations (black), monthly mean modeled concentrations (red), monthly RMSE (blue), monthly spatiotemporal correlations (green), monyhly MFB (cyan) and monthly MFE (magenta). Solid curves refer to the left axis while dotted curves refer to the right axis.



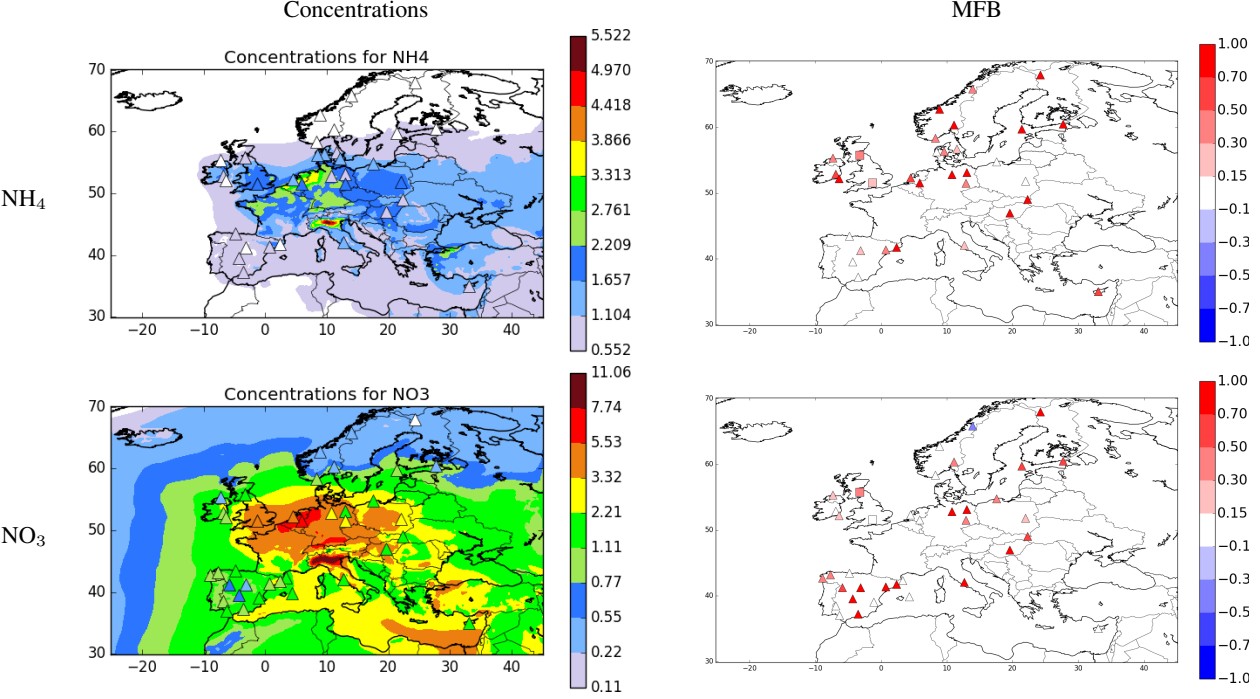

**Figure 6.** Modeled concentrations (in $\mu$g m$^{-3}$) and MFB for NH$_4$ and NO$_3$ in 2013. Triangles correspond to measured concentrations in the left and to the MFB value in the right.



**Figure 7.** Seasonal evolution of statistics by regions for NO$_3$: Monthly mean measured concentrations (black), monthly mean modeled concentrations (red), monthly RMSE (blue), monthly spatiotemporal correlations (green), monyhly MFB (cyan) and monthly MFE (magenta). Solid curves refer to the left axis while dotted curves refer to the right axis.



**Figure 8.** Seasonal evolution of statistics by regions for $NH_4$: Monthly mean measured concentrations (black), monthly mean modeled concentrations (red), monthly RMSE (blue), monthly spatiotemporal correlations (green), monyhly MFB (cyan) and monthly MFE (magenta). Solid curves refer to the left axis while dotted curves refer to the right axis.



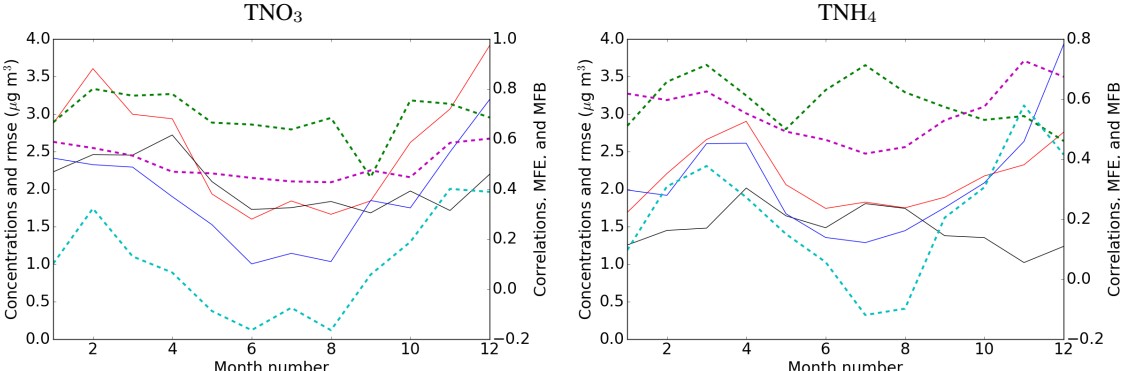

**Figure 9.** Seasonal evolution of statistics for the all Europe for TNO$_3$ and TNH$_4$: Monthly mean measured concentrations (black), monthly mean modeled concentrations (red), monthly RMSE (blue), monthly spatiotemporal correlations (green), monthly MFB (cyan) and monthly MFE (magenta).

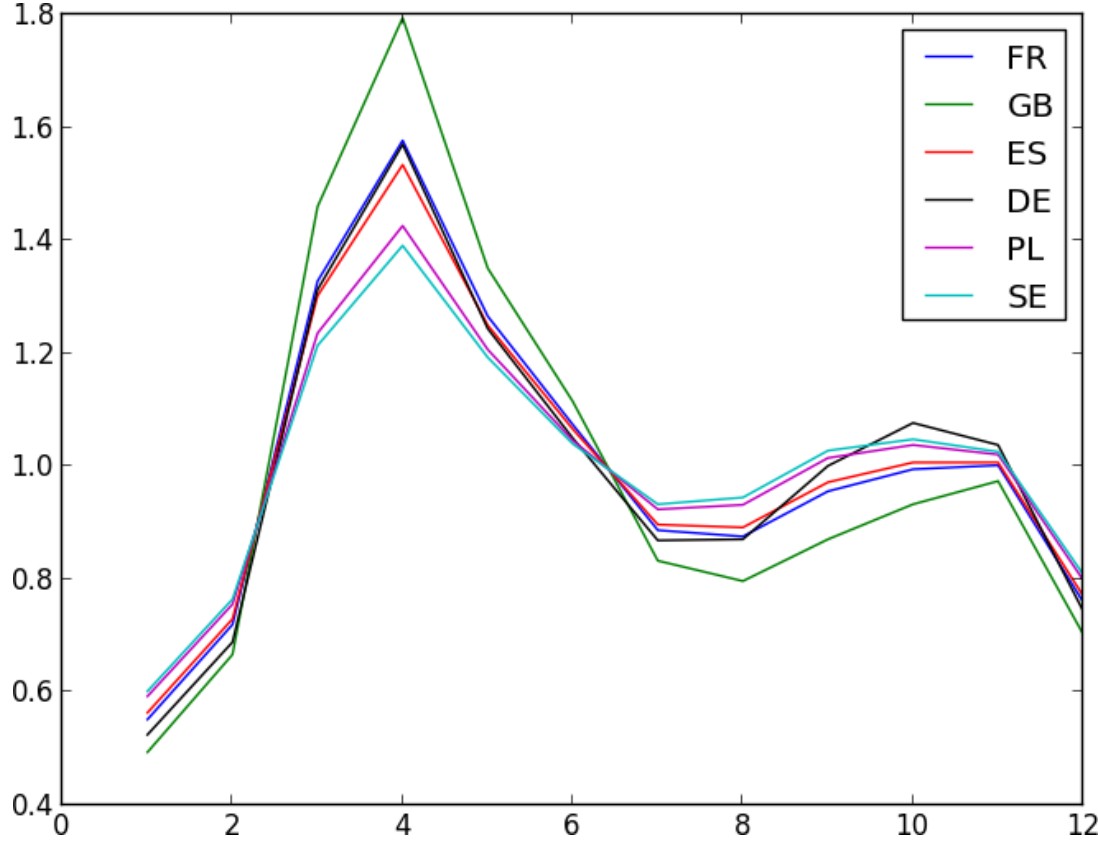

**Figure 10.** Seasonal evolution of NH$_3$ emissions used in CHIMERE for several countries.





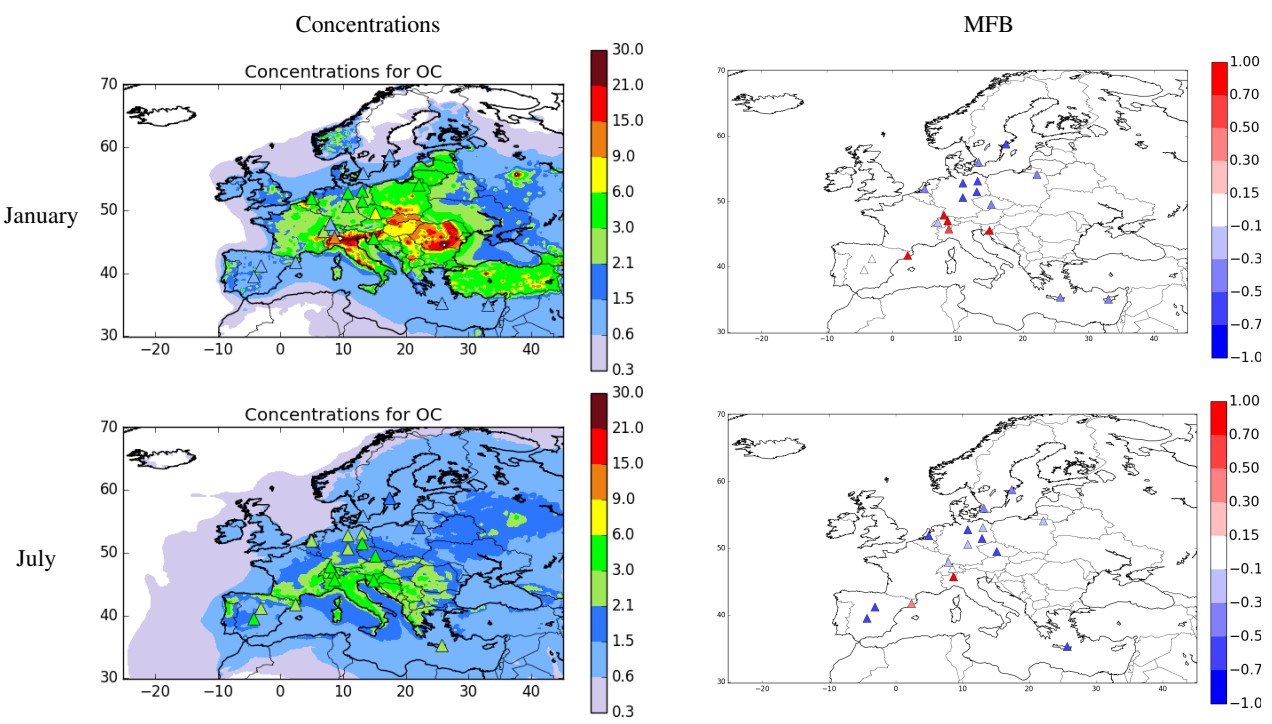

**Figure 11.** Modeled concentrations (in $\mu$g m$^{-3}$) and MFB for Organic Carbon in January and July 2013. Triangles correspond to measured concentrations in the left and to the MFB value in the right.





**Figure 12.** Temporal evolution of modeled (red line) and measured (black line) OC concentrations (in in $\mu$g m$^{-3}$) for stations in the South of Europe. The green line corresponds to Organic Carbon from biogenic compounds.





**Figure 13.** Temporal evolution of modeled (red line) and measured (black line) Organic Carbon concentrations (in in $\mu$g m$^{-3}$) for stations in the South of Europe. The green line corresponds to OC from biogenic compounds.





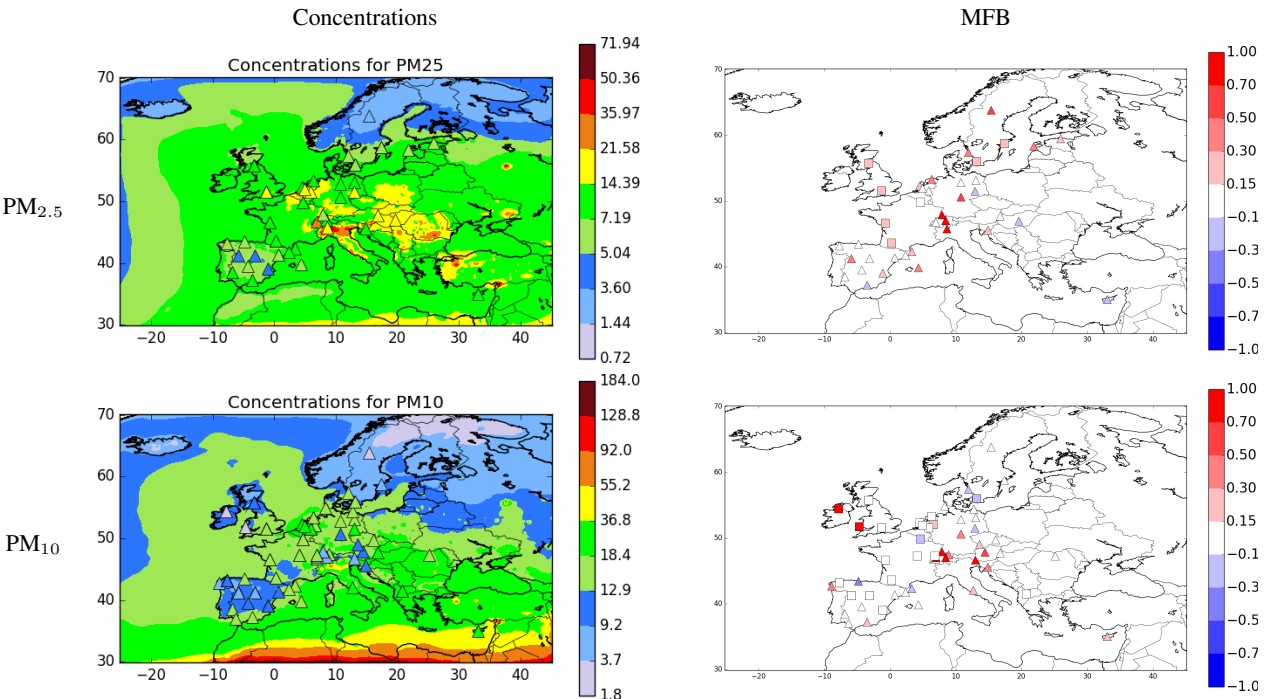

**Figure 14.** Modeled concentrations (in in $\mu$g m$^{-3}$) and MFB for PM2.5 and PM$_{10}$ in 2013. Triangles correspond to measured concentrations in the left and to the MFB value in the right.



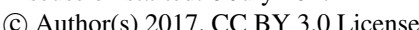

**Figure 15.** Seasonal evolution of statistics by regions for PM$_{2.5}$: Monthly mean measured concentrations (black), monthly mean modeled concentrations (red), monthly RMSE (blue), monthly spatiotemporal correlations (green), monyhly MFB (cyan) and monthly MFE (magenta). Solid curves refer to the left axis while dotted curves refer to the right axis.





**Figure 16.** QQ scatter plot of $PM_{10}$ and of $PM_{2.5}$ (bottom right panel) modeling results against measurements for the several regions. SE: Southern Europe. WE: Western Europe. CE: Central Europe. EE: Eastern Europe. NE: Northern Europe.



**Figure 17.** Seasonal evolution of statistics by regions for PM$_{10}$: Monthly mean measured concentrations (black), monthly mean modeled concentrations (red), monthly RMSE (blue), monthly spatiotemporal correlations (green), monyhly MFB (cyan) and monthly MFE (magenta). Solid curves refer to the left axis while dotted curves refer to the right axis.

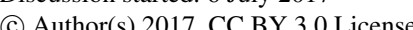



**Figure 18.** Modeled (red) and measured (black) concentrations of $PM_{2.5}$, $PM_{10}$, Ca (fine fraction and total) and $NO_3$ (fine fraction and total) in $\mu$g m$^{-3}$ for the Cyprus station CY0002R.



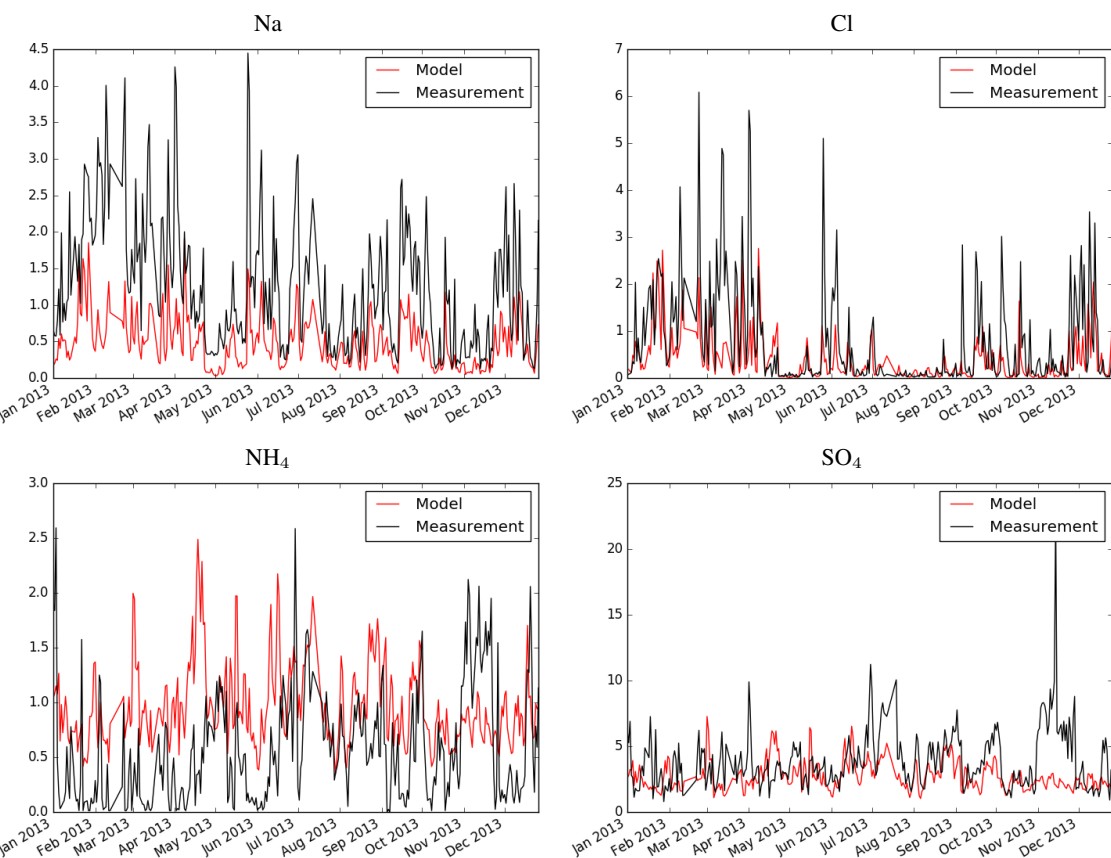

**Figure 19.** Modeled (red) and measured (black) concentrations of Na, Cl, $NH_4$, $SO_4$ in $\mu$g m$^{-3}$ for the Cyprus station CY0002R.







**Figure 20.** Modeled (red) and measured (black) concentrations of PM$_{2.5}$, PM$_{10}$, Ca (fine fraction and total) and NO$_3$ (fine fraction and total) in $\mu$g m$^{-3}$ for the station DE0044R (Germany)



**Figure 21.** Modeled (red) and measured (black) concentrations of Na, Cl, NH$_4$, SO$_4$, EC in $\mu$g m$^{-3}$ for the station DE0044R (Germany).