# Peer review of "Development of an inorganic and organic aerosol model (CHIMERE 2017 $\beta$ v1.0): seasonal and spatial evaluation over Europe"

_Geoscientific Model Development, 2017_

## Short Comment (SC1) · 12 Jul 2017

Dear authors,

In my role as Executive editor of GMD, I would like to bring to your attention our Editorial version 1.1:

http://www.geosci-model-dev.net/8/3487/2015/gmd-8-3487-2015.html

This highlights some requirements of papers published in GMD, which is also available on the GMD website in the 'Manuscript Types' section:

http://www.geoscientific-model-development.net/submission/manuscript_types.html

In particular, please note that for your paper, the following requirements have not been met in the Discussions paper:

- "All papers must include a section, at the end of the paper, entitled 'Code availability'. Here, either instructions for obtaining the code, or the reasons why the code is not available should be clearly stated. It is preferred for the code to be uploaded as a supplement or to be made available at a data repository with an associated DOI (digital object identifier) for the exact model version described in the paper. Alternatively, for established models, there may be an existing means of accessing the code through a particular system. In this case, there must exist a means of permanently accessing the precise model version described in the paper. In some cases, authors may prefer to put models on their own website, or to act as a point of contact for obtaining the code. Given the impermanence of websites and email addresses, this is not encouraged, and authors should consider improving the availability with a more permanent arrangement. After the paper is accepted the model archive should be updated to include a link to the GMD paper."

Therefore please provide a reason why CHIMERE is not publicly available. If it is publicly available, please make the exact version your article refers to available via a permanent archive providing a DOI (e.g. Zenodo).

Yours,

Astrid Kerkweg
* * *

---

## Referee Comment (RC1) · Anonymous Referee #1 · 2 Aug 2017

The paper describes a series of developments in the chemical transport model Chimere and the evaluation of the model against aerosol measurements in several stations in Europe during 2013. The changes include new biogenic VOC emissions, replacements of the inorganic aerosol model ISORROPIA with that of ISORROPIA-II, update of the secondary organic aerosol module and better description of wet deposition. The description of the changes and the evaluation of the model are satisfactory. The scientific content of the paper could be improved by better connection of the changes made with the predictions and also the performance of the model. Overall, the paper is suitable for publication in GMD after some improvements and corrections are made. These are described below.

[Figure]

**General Comments**

(1) One of the major weaknesses of the work is that there is very little about the effect of the various changes on the predictions of the CTM. I understand that repeating these simulations with the previous version of Chimere requires significant work and may not be worthwhile. However, some discussion of the results of previous evaluations of the model is necessary.

(2) Abstract. The use of the terms "good performance", "good seasonal evolution", "performs well", etc., should be avoided or at least the quantitative metrics should be presented instead.

(3) Abstract and rest of the paper. Most of the work focuses on PM10 with some analysis performed for PM2.5. However, it is not clear in most of the paper if the concentrations refer to the former or the latter. I think that the size-range should be mentioned everywhere in the paper to avoid confusion.

(4) Abstract. The conclusion that the model "strongly underestimates SOA concentrations" is not supported by the evaluation which is based on the comparison with total OC concentrations. The same applies to the explanation that "this underestimation could be due to a lack of anthropogenic SOA in the model".

**Model description**

(5) It is not clear if the composition of dust is taken into account in the calculation of cloud pH.

(6) The volatilities of the three POA components should be mentioned. Also despite the corresponding discussion it is not clear to me how their emissions are calculated from the OA emissions in the corresponding inventory for the various sources. A table with the corresponding volatility-resolved emission rates for the various sources would be helpful.

(7) The POA aging reactions (1)-(3) are a net sink of OH. In the rest of the SOA reactions it is assumed that the OH is recycled. Why are these treated differently?

(8) If my understanding is correct the authors assume that the primary OA emissions in the inventory that they are using represent 20 percent of the organic compounds emitted in a certain volatility range (less or equal than 100 $\mu$g m$^{-3}$ as IVOCs are neglected). So the corresponding emissions are multiplied by a factor of 5. Given that a lot of these emissions have been measured at high OA levels this increase appears to be too high. It appears that the emissions used set the IVOC emissions to zero and at the same time add a lot of organic emissions to the more volatile SVOCs. This is an important issue for the model so some additional analysis of what exactly is done and why is necessary.

(9) The use of ISORROPIA-II is rather confusing. Its major advantage compared to ISORROPIA is its ability to simulate the thermodynamics of calcium, magnesium, etc., compounds. However, the authors state in Section 2.1.4 (page 5, lines 22-24) that the crustal elements are not taken into account for the partitioning. This counterintuitive choice requires some justification.

(10) Some discussion of the aerosol size resolution used in the model is needed.

(11) Section 2.1.8. The way that the mass transfer is between the gas and particulate phases is simulated is not entirely clear. If my understanding is correct, bulk equilibrium is assumed for both inorganics and organics for sizes below 1.25 micrometers. However, it is not clear what happens for coarse particles for compounds like ammonia, hydrochloric acid, organics, etc. Also is the formation of calcium nitrate the only way that nitrate can be transferred to the coarse particles? What about sodium nitrate?

(12) Tables 1-3. A list of the species names probably in the Supplementary Information is needed to understand the information in these tables.

(13) The Henry's law constants of a number of organic species are both in Tables 2

and 4 and in quite different units. These important parameters should be in one table with some explanation about their source. Also the fact that they appear to be known quite accurately (three significant digits in Table 2) is problematic.

**Emissions**

(14) The authors mention that they use the EMEP inventory published in 2003. Are they using any emissions updates for 2013 or the emissions used are that old? A table summarizing the anthropogenic emissions used for each major source category would be helpful together with additional information about the origins of these estimates. The biomass emission rates should be mentioned.

(15) How are the sea-salt emissions calculated? What is the assumed composition and size distribution? This scheme appears to perform quite well based on the Chimere evaluation presented here.

(16) Some additional discussion of the ammonia emissions used would be helpful. Is there a diurnal variation of the emissions? Is the monthly variation (shown in Fig. 10 for six countries) similar for all areas in the domain? What are the total emissions?

**Observations**

(17) A short description of the various stations used in the paper is needed. This could be part of the Supplementary Information. My understanding is that all of them are regional background stations. Is this correct? Are there any exceptions?

(18) There appear to be around 20 stations measuring sodium in Figure 2. However, 38 stations are mentioned in Table 5. What is causing this discrepancy? There are similar issues with other PM components.

(19) A number of urban background stations (e.g., in Paris) could have also been used in the evaluation. Is there a reason why they have been excluded? Also there should

be PM1 measurements from the ACTRIS network available for the same period.

(20) How well did the measured PM mass concentrations compare to the sum of the components? Could some of the measured PM mass be water as previous studies have indicated for Europe but also for the US?

(21) The number of measurements used in the evaluation should be mentioned in Table 5.

**Boundary conditions**

(22) A brief summary of the boundary condition values (or ranges) provided by Mozart for the various sides of the domain would be helpful. It appears that these boundary conditions may be partially responsible for some of the discrepancies between model predictions and observations in areas like Northern Europe.

**Carbonaceous aerosol**

(23) The authors discuss the potential errors in the comparison of model-predicted OM values with OC measurements. However, they never discuss the actual values of the OM/OC ratios that they use. I understand that these vary temporally, but the average values for each station for at least the summer and winter could be presented. Given that there are now a lot of measurements of this ratio by High-Resolution Aerosol Mass Spectrometers in locations around Europe a more informed comparison could be made.

(24) The study focuses on daily average measurements. However, some discussion of the diurnal variations of components like OM would have been welcome. There are measurements of these variations in stations like Melpitz that could be useful.

(25) The predicted high levels of OC in parts of central and eastern Europe during the winter appear to be too high. I am assuming that these are due to biomass burning.

Is this an indication of emissions that are too high? May be atmospheric mixing that is too weak? Some discussion is needed here because other authors have argued that the biomass burning emissions in the European inventory is too low. Could it also be a problem due to the high SVOC emissions used by the model that become particles under low temperatures?

(26) There is no discussion regarding the predictions of EC by Chimere.

**Nitrates**

(27) Some discussion of the evaluation of coarse nitrate predictions by Chimere are needed at least in Cyprus but also in other areas where PM2.5 and PM10 nitrate measurements may be available. A plot of this coarse nitrate in Figure 18 would be useful. Could some of the nitrate prediction problems be due to the challenges in predicting coarse nitrate?

(28) The model predicts high nitrate levels in the southeastern Mediterranean (Figure 6). This is rather unexpected due to the relatively low NOx levels and high sulfate concentrations. The values appear to be quite high compared to what has been observed (there are available measurements in Crete). Is this due to dust? How does the model produce so much nitric acid in that area?

**Figures**

(29) Figures 3, 5, 7, 8, and 9 are quite confusing because they contain too much information and do not have legends. One solution would be to show just the predicted and measured concentrations in these figures and then show the evaluation metrics in the supplementary information. It would be nice to also indicate the number of stations in each area in these figures.

(30) The axes of Figure 10 do not have titles.

(31) There are several cases with missing data in the time series of Figures 12, 13, etc. These are replaced by lines connecting the existing measurements. This is confusing and may be misleading. There should be gaps in the corresponding lines. Even better symbols could be used instead of lines for the measurements.

(32) An explanation of what is shown in the QQ scatter plot (Figure 16) is needed.

**Other issues**

(33) The use of Na instead of $Na^+$, $SO_4$ instead of $SO_4^{2-}$, etc., throughout the manuscript is problematic and should be avoided.

(34) There are a number of typos that should be corrected. Some of them:

Page 7, line 18. NH3

Page 7, line 26, Henr's

Page 11, lines 29-20, with strong observed of Na

Page 20, line 24, S. et al. (2016)

Page 26, lines 4-5. Missing author names.

Figure 3, 5, 7, 8 , 9 captions. monyhly.

Page 46, line 1. PM2.5
* * *

---

## Referee Comment (RC2) · Anonymous Referee #2 · 2 Aug 2017

The manuscript describes an updated aerosol module for the CHIMERE regional air quality model, along with evaluation of these changes against surface concentration measurements over Europe. The updates cover a number of different processes within the model: emissions, wet deposition, evaporation and condensation of both organic and inorganic semi-volatile components and hygroscopic growth. A set of performance criteria from the literature are adopted and the model is shown to perform well against these.

However, while the work itself is a worthy contribution to the field of aerosol modelling for air quality applications, there are a number of deficiencies in the presentation such

that I would recommend major revisions before the manuscript is suitable for publication in GMD.

**General comments**

1. There are a very large number of figures (21), many of them with multiple panels and similar in nature and with dense high-frequency time series that are hard to interpret. This makes it difficult for the reader to discern what are the important results being presented. If this level of detail is necessary for completeness, it would be better placed in supplementary material, and a smaller number of clearer figures used to (i) exemplify the raw data, and (ii) summarise its meaning statistically in a visual form.

2. The manuscript presents an updated version of an existing model; however it is frequently unclear how the new schemes described here compare to those used in the reference/baseline version of CHIMERE (Menut et al., 2013) in their formulation and complexity. Corresponding results for the reference version should also be included, in order to assess not only the absolute performance of the revised model, but to what extent the changes described in this manuscript produce improvements in these performance metrics.

3. In several places in the manuscript, positive and negative biases and larger or smaller errors and correlations are shown at individual stations and over various regions. These may very well be statistically significant variations, however the analysis presented does not adequately demonstrate this.

**[GMDD](https://www.geosci-model-dev-discuss.net/)**
**Specific comments**

4. Page 1, line 1. This describes a "new" aerosol module, although elsewhere it is clear that this is in fact an update to an existing module; the introductory text should be re-worded accordingly.

5. Page 1, lines 1–17. It would be good to see some quantitative results quoted in the abstract about the performance of the updated model and how that compares to the reference/baseline version.

6. Page 1, line 19–page 2, line 3. The introductory paragraph is quite vague on the subject of why such models are useful, despite the list of model references. A little more background on the motivation for such modelling would be welcome.

7. Page 2, line 10. What definition of "fine" is being used in this context?

8. Page 3, line 19–21. An overview of the reference/baseline model version here would be very useful – overall approach and assumptions, what are the tracers used, does it represent the particle size distribution or is it a bulk scheme etc.? This would also make it easier to clarify in the rest of the section how the updated schemes relate to this baseline.

9. Page 3, line 24–page 4, line 10. How does this compare to the chemical mechanism in the baseline version?

10. Page 4, lines 13–22. How does this compare to the treatment of biogenic emissions in the baseline version?

11. Page 4, line 25. A brief discussion of what these emissions are would be helpful, even if further detail is to be found in the reference.

[Figure]

12. Page 4, line 25–page 5, line 17. This subsection cites various conflicting studies, but leaves the reader unclear as to what conclusion is drawn for the purposes of this work.

13. Page 5, line 20–page 6, line 14. How does this compare to the treatment of aerosol thermodynamics in the baseline version?

14. Page 6, line 10. A description and/or reference should be provided for the "$H_2O$ mechanism".

15. Page 8, line 1–page 9, line 1. How does this compare to the treatment of wet deposition in the baseline version?

16. Page 9, lines 3–28. How does this compare to the treatment of condensation/evaporation in the baseline version?

17. Page 10, lines 1–8. How does this compare to the treatment of coagulation in the baseline version?

18. Page 10, lines 11–12. There are several IFS-based products from ECMWF. Please clarify whether this refers to operational analyses or forecasts, or to one of one of the reanalyses (e.g. ERA-Interim).

19. Page 11, lines 7–11. The description of the observations is very brief, and would benefit from being extended – e.g. what type of instruments to these measurements come from, and how extensive is its coverage in space and time? Also, a reference and acronym expansion should be provided for EBAS if possible.

20. Page 11, lines 23–24. Please explain why this is a likely explanation for the $Na$ and $Cl$ results.

21. Page 13, lines 23–25. It could also be that a third factor which is poorly captured in the model affects both sulfates and nitrates.

22. Page 14, line 20. Please describe and/or give a reference for MELCHIOR 2.

23. Page 15, line 9. It is not clear whether "OC concentrations were calculated directly" from the model or from observations. Please clarify.

24. Page 16, line 6. MFB for $PM_{10}$ is still positive, suggesting coarse particles are still *over*estimated, just less so than smaller particles.

25. Page 17, line 26. This sounds like the *measurements* are overestimated, but presumably is intended to say that the *model* overestimates $NO_3$ *compared to* the measurements?

26. Page 17, line 34–page 18, line 1. Please explain why a lack of $HNO_3$ condensation is likely to explain this.

27. Page 20, line 23. A reference to the data referred to here would be good.

**Technical corrections**

28. Page 1, line 8 (and elsewhere): Performances were $\longrightarrow$ Performance was.

29. Page 1, lines 8 and 10 (and elsewhere): sea salts $\longrightarrow$ sea salt.

30. Page 1, line 15: most of stations $\longrightarrow$ most of the stations.

31. Page 2, line 18: dusts $\longrightarrow$ dust.

32. Page 2, line 24: aerosols thermodynamics $\longrightarrow$ aerosol thermodynamics.

33. Page 3, line 15: described in *which* part of the paper?

34. Page 3, lines 25–26: "a" should be before "function", not "partitioning".

35. Page 3, line 28: insert "and" before "equilibrium constants".

36. Page 4, line 14: "temperature" and "solar" should not be capitalised.

37. Page 4, lines 16–17: "wilting point" should not be capitalised.

38. Page 4, line 20: landuse $\longrightarrow$ land-use.

39. Page 5, line 14: vehicle $\longrightarrow$ vehicles.

40. Page 5, line 15: aromatics compounds $\longrightarrow$ aromatic compounds.

41. Page 9, line 13: $Delta \longrightarrow \Delta$.

42. Page 9, lines 15 and 17: what is $Kn$?

43. Page 9, line 21: insert "in" before "computed with".

44. Page 9, line 25: $CaCO_3$ of dusts $\longrightarrow$ $CaCO_3$ in dust.

45. Page 10, lines 12 and 15: "temperature" should not be capitalised.

46. Page 10, line 15: "wind speed" should not be capitalised.

47. Page 10, line 16: underestimation of the PBL around... $\longrightarrow$ underestimation of the PBL height of around...

48. Page 10, lines 17–18: "boundary conditions" should not be capitalised.

49. Page 10, lines 21–22 (and elsewhere): performances $\longrightarrow$ performance.

50. Page 11, line 25: than $\longrightarrow$ as.

51. Page 11, lines 29–30: observed of $Na$ $\longrightarrow$ observed $Na$.

52. Page 14, line 3: "summer" should not be capitalised.

53. Page 15, line 19: criteria is ⟶ criteria are.

54. Page 15, lines 19–20: overestimation...do not correspond ⟶ overestimation...does not correspond.

55. Page 16, line 18: delete "the" before "Southern Europe".

56. Page 16, line 24: $PM_{2.5}$ are ⟶ $PM_{2.5}$ is.

57. Page 16, lines 25, 31–32: "winter", "spring", "summer" and "fall' should not be capitalised. Also, please use either "fall" or "autumn" consistently throughout – both appear in the manuscript.

58. Page 16, line 32: due to mostly to ⟶ due mostly to.

59. Page 17, lines 20–23. This sentence is confusing, with two consecutive "but" clauses and multiple parentheses. Consider breaking it up to clarify the meaning.

60. Page 17, line 27: faction ⟶ fraction.

61. Page 18, line 14: "April" and "August" should be capitalised.

62. Page 18, line 31: could be explain ⟶ could be explained.

63. Page 19, line 4: emissions is ⟶ emissions are.

64. Page 20, line 13: inorganics aerosol ⟶ inorganic aerosol.

65. Page 20, line 24: the "S. et al" citation should have a full surname, not just an initial. (The same applies to all authors in the corresponding bibliography entry.)

66. Page 21, line 9: dynamic ⟶ dynamics.

67. Page 21, line 14: "CHIMERE" should be capitalised as elsewhere in the manuscript.

---

## Referee Comment (RC3) · Anonymous Referee #3 · 22 Aug 2017

This manuscript presents an update of the CHIMERE's aerosol module. The chemical mechanism was modified for the formation of the secondary organic aerosol precursors. The equilibrium between the aerosol and the gas phase is then treated using the module SOAP. For the secondary inorganic aerosols, the thermodynamic equilibrium is computed using ISORROPIA, which has been updated to the version 2.1 here. Biogenic emissions have been updated and are now computed using MEGAN 2.1. Below-cloud scavenging has also been updated. After a presentation of the model CHIMERE and the aerosol module, the developments are validated over the year 2013 using surface measurements from the EBAS database by separating Europe into 5 coherent sub-regions based on country borders. The authors give some recommendations for

future development. Finally the manuscript ends with a conclusion.

This manuscript is interesting for the aerosol community as it presents recent features in aerosol modelling. However, it presents some serious issues for a publication as it is.

The manuscript presents a new aerosol module (cf. abstract), but otherwise in the text it is treated as an update of the existing module. It is sometimes difficult to differentiate the parametrization and developments related to the 2013 version from the new 2017 version. The authors need to be clearer about this point that highlights the important development work done here. I would recommend to present the 2013 version of the parametrizations before introducing the new features. Following this remark, there is no comparison between the 2013 and 2017 versions over the year 2013. It would be very interesting to see the evolution in the performance of the model between the two versions.

The comparison to the observation set is very interesting as it uses a lot of collocated information on several measuring stations. It is then possible to evaluate the aerosol load, but also the aerosol composition and the seasonality. All these pieces of information could point out easily the strengths and the weaknesses of the model. However the authors only use ground based stations. It would have been interesting to compare the simulation to vertically integrated measurements such as aerosol optical depths, especially to evaluate the impact of the changes on wet deposition.

Also, almost all the mathematical formulations need to be reviewed. There are for example undefined variables used in equations or discrepancies between the name of a variable in the text and in an equation.

I would then recommend major revisions before publication.

**General comments:**

1. The use of paragraph breaks is sometimes puzzling, for example on page 13 line

19. This sentence might make think that the next paragraph does not refer to the figures 7 and 8, which is not the case.

2. Page 2 line 2: I do not understand the presence of the "Vestreng, 2003" reference for the air quality model.

3. Section 2.1.1: Maybe a table that summarizes the reaction rate for the oxidation of $SO_2$ in clouds could be useful.

4. Section 2.1.3: I didn't understand how are the anthropogenic emissions managed for the POA and SVOC. You use POA emissions from an emission inventory. These POA are emitted into the species POAlP, POAmP and POAhP. Then you use the quantity of POA emitted to emit the SVOC by saying that SVOC = 5 * POA and say that you don't take into account the IVOC. Is this right? How are then the SVOC emitted into the MELCHIOR species?

5. Page 6 line 4: I understand the dynamic method requires a lot of computation time explaining you choices. But did you run a test to know the impact it could have on a specific test-case for example?

6. Page 7, line 1: pleas also add "the mass fraction of respectively the solid phase, the aqueous phase and the organic phase **in the particle**" for a better understanding.

7. Section 2.1.6: The title is "Dry deposition of particles and semi-volatile organic species" but you talk about other gaseous species such as $O_3$, $SO_2$, etc. Please change the text to be consistent.

8. Section 2.2: When talking about the simulation set-up, you don't talk about the vertical resolution of the simulation made. Please add these informations.

9. Page 10, line 18: You wrote "Boundary Conditions were generated from [...]". Could you explain how they were generated?

10. Section 2.3: This part is too small. Could you please add a table with the total number of stations and the number of stations in each of the regions you defined. According to Fig. 1 there are no stations for western Europe. Could you please explain why? Also, please add the link to the EBAS database in the text.

11. Page 11, line 19: You wrote "Only one station in Spain underestimates [...] concentrations for Na." while on Fig. 1 there are three blue triangles.

12. Page 11, line 29: You talk about the station ES0008R, but without showing any figures. Also this sentence is hard to understand.

13. Page 12 line 2: "The other stations [...] the opposite trend". I do not understand which other stations? All the stations or the other stations from southern Europe? Could you please make this part clearer?

14. Section 3.2: You are talking about sulfate observations. But are you using "total sulfate" or "corrected sulfate" (or non sea salt sulfate) measurements?

15. Page 14, line 23: I do not understand how an overestimation of NH3 emissions could induce an overestimation of TNO3. Could you please explain it with more details?

16. Section 3.3.1: How many measurements do you have for each stations in Table 6? it seems that some stations have very few measurements points, e.g. CH0002R looks like to have around 10 points. What is the confidence in the statistics you can have in this case?

17. Page 16, line 16: "especially in the Alps". How is the relief represented in the model in this area? Can it explain the overestimation of the PM by the model? Maybe you could try to interpolate the model at the altitude of the stations to improve the comparison made.

18. Page, line 30: What is the effect of the strong overestimation over the Alps on the results of Fig. 15 and 17 for central Europe?

19. Section 3.3.3: You talk about Ca and $NO_3$ in $PM_{10}$, but in Fig. 18 there is "total Ca" and "total $NO_3$". Are these two quantities supposed to be the same? Please change the text to be clearer.

**Figure related comments:**

1. The manuscript preparation guidelines for authors claims that: The abbreviation "Fig." should be used when it appears in running text and should be followed by a number unless it comes at the beginning of a sentence, e.g.: "The results are depicted in Fig. 5. Figure 9 reveals that...". Please check that the right word is used in the text.

2. Also the authors should check the legend of their figures that are not always satisfactory. For example, the legend of the first figure does not mention what are the measuring station marked by dots.

3. Concerning the time series, it seems that there are missing values in the observations linked by a segment in the figures (e.g. On Fig. 12 between Feb and Mar 2013 for GR0002R). This can be misleading for the interpretation of the figure. Could you please change this? Maybe you could use symbol for the measurements and have continuous line for the simulation results.

4. Figure 2, 4, 6, 11 and 14: These figures does not seem to be complete on the right side for negative numbers. What represent the squares on these figures?

5. Figure 3, 5, 7, 8, 9, 15, 17: Is it possible to have slightly thicker lines? Is it possible to draw a line for 0 in order to read more easily some quantities such as the MFB.

[Figure]

6. Figure 3: All your sub figures does not have the same size or position.

7. Figure 12 and 13: All your sub figures does not have the same size.

8. Figure 13: There seems to be a mistake. I think "CH0005R" should be "CY0002R".

9. Figure 16: The figure for $PM_{2.5}$ is very small. Is it possible to enlarge it?

**Table related comments:**

1. Table 2, line "BiDer": hydrohilic -> hydro**p**hilic.

2. Table 2: for the type you mention type A, B and C, but in the table you only write hydrophilic or hydrophobic. Please explain this a little more.

3. Table 4: What are the Henry's law constant used for the other species such as $HNO_3$?

4. Page 14, line 8: Please add a reference to Table 5 when talking about TNO3 and TNH4.

5. Table 5 presents results for $O_3$ and $NO_2$ but you never talk about them in the text.

**Technical comments** (when a letter or a word is missing, it is in bold in the comment):

1. Page 3, line 15: I guess the word "first" is missing in the sentence.

2. Page 5, line 19: "Thermodynamic of Secondary organic and [...]" -> "Thermodynamic of secondary organic and [...]"
3. Section 2.1.5: the symbols $w$ are different but refers to the same quantity. So are $d$ and d. There is also a discrepancy between $d_{sol}$ and $d_{solid}$.

4. Page 7, line 9-12: $F_d, i$ is different from $\mathsf{F}_{dry,i}$. Same for $v_{d,i}$, $\mathsf{v}_d$ and $\mathsf{v}_d, i$. What represents $C_i$?

5. Page 7, line 25: $\rho_{eau}$ and $\rho$. What is $\mathsf{M}_{eau}$? Does $H_i$ stand for the Henry's law constant?

6. Page 7, line 26: Henr**y**'s law.

7. Page 9, line 12: there is no verb in the sentence.

8. Page 9, Eq. 18: What is $P$?

9. Page 9, Eq. 19: $Delta$ -> $\Delta$, $k^{bin}$ -> $k_i^{bin}$

10. page 9, Eq. 20: What is $R$ and $T$?

11. Page 9, Eq. 21: $Delta$ -> $\Delta$

12. Page 10, line 2: $J_{coag,i}^{bin}$ and $J_{coag,i}^{b}$

13. page 10, Eq. 23: $K_{j,k}$ instead of $K_{j,l}$. What is $A_{p,i}$? The second sum symbol does not have attribute. Is the sum going from 1 to b over j?.

14. Page 10, line 13: Wind Speed -> wind speed (also line 15).

15. Page 10, line 13: Planetary Boundary Layer (PBL) -> planetary boundary layer (PBL).

16. Page 10, line 15: Temperature -> temperature.

17. Page 10, line 18: Mozart -> MOZART.

18. Page 10, line 18: "Boundary Conditions" -> "boundary conditions"

19. Page 11, line 5: Figure 1 -> Fig. 1.

20. Page 12 line 17: Tsyro et al. (2011) **and** Neumann et al. (2016).

21. Page 15, line 11:"OM/OC simulated" -> "OM/OC ratio simulated".

22. Page 15, line 29: "summer concentrations are underestimated in summer". Summer is written twice.

23. Page 16, line 18: "PM2.5 and PM10, respectively" -> "PM2.5 and PM10 respectively", no comma.

24. Page 18, line 9: "Figure 12" -> "Fig. 13".

25. Page 19, line 9: "to take into IVOC emissions", it seems that a word is missing here.

---

## Author Comment (AC1) · 26 Oct 2017

We would like to thank all the reviewer for the interesting comments. Before addressing the specific comments of all the reviewer, we would like to address the common comment on the comparison between the CHIMERE 2013 and CHIMERE 2017β versions. One of the reason that originally motivated this study is the lack of numerical stability of the model in some configurations (illustrated by the following figure obtained over a station), but also problems in the representation of several processes in the model (like coagulation which was based on an old parameterization not taking into account the number of particles or condensation/evaporation routines which has numerical issues and did not represent to our opinion adequately the process). Moreover, a few bugs were present in the Chimere2013 version. The improvements referred inside the text concern all these issues.

[Figure]

Due to this, the aerosol modules (except for the nucleation subroutine) was entirely rewritten.

Comparing the two versions does not have to our opinion any sense as Chimere2013 is not stable in the closest configuration possible. The aim of this study is not to change a few parameterizations and to see their effect but to present the results of a new module. We chose therefore to present the results of this version without referring to the Chimere 2013 because:

1. It is almost impossible to run the Chimere 2013 version without changing the configuration and the code
2. Even if it was the case, differences would be difficult to analyze because of some corrections and bugs present in Chimere 2013.

The aim of this study was therefore only to present the new module without referring on how these parameterizations differ from Chimere 2013. Presenting the differences between the two versions was done in the first version of the paper and it was found to be mainly confusing and deserving the quality of the paper. Moreover, presenting the changes between the Chimere 2013 and Chimere 2017beta versions gave the wrong idea that only a few modifications were done, whereas it is indeed a new module. A few references to the Chimere 2013 were however still present and may have been confusing. The references to the Chimere 2013 version were limited and the text was modified to emphasize the fact the results of a new aerosol module is presented. However, a table presenting the differences between the two versions is added in Supplementary Materials.

*Anonymous Referee #1*

*The paper describes a series of developments in the chemical transport model Chimere and the evaluation of the model against aerosol measurements in several stations in Europe during 2013. The changes include new biogenic VOC emissions, replacements of the inorganic aerosol model ISORROPIA with that of ISORROPIA-II, update of the secondary organic aerosol module and better description of wet deposition. The description of the changes and the evaluation of the model are satisfactory. The scientific content of the paper could be improved by better connection of the changes made with the predictions and also the performance of the model. Overall, the paper is suitable for publication in GMD after some improvements and corrections are made. These are described below.*

*General Comments*

*(1) One of the major weaknesses of the work is that there is very little about the effect of the various changes on the predictions of the CTM. I understand that repeating these simulations with the previous version of Chimere requires significant work and may not be worthwhile. However, some discussion of the results of previous evaluations of the model is necessary.*

Addressed in the general reply.

*(2) Abstract. The use of the terms "good performance", "good seasonal evolution", "performs well", etc., should be avoided or at least the quantitative metrics should be presented instead.*

The abstract was modified.

*(3) Abstract and rest of the paper. Most of the work focuses on PM10 with some analysis performed for PM2.5. However, it is not clear in most of the paper if the concentrations refer to the former or the latter. I think that the size-range should be mentioned everywhere in the paper to avoid confusion.*

To prevent weighting down the paper by specifying every time the fraction, the following sentence was added at the beginning of the section "Results":

"When not mentioned otherwise, concentrations of components used for the comparison are in the PM10 fraction."

*(4) Abstract. The conclusion that the model "strongly underestimates SOA concentrations" is not supported by the evaluation which is based on the comparison with total OC concentrations. The same applies to the explanation that "this underestimation could be due to a lack of anthropogenic SOA in the model*

"strongly underestimates SOA concentrations". replaced by "strongly underestimates summer organic aerosol concentrations". More discussions were added into the text and the conclusions was nuanced by arguing that the underestimation could also be due to a lack of biogenic emissions

"For the northern half of Europe, except for stations DE0003, CH0005R in Switzerland and PL0005R in Poland which have strong concentrations of modeled biogenic SOA, summer concentrations of organic aerosol are underestimated. Only a peak of organic aerosol (due to biogenic aerosols in the model) at the end of August for several stations (CZ0003R, DE0002R, DE0007R, DE0008R, DE0044R) is reproduced by the model. These stations correspond to areas with strong anthropogenic emissions. A lack of anthropogenic SOA could therefore explain this pattern. However, this underestimation could be also due to a lack of biogenic emissions over these areas."

*Model description*

*(5) It is not clear if the composition of dust is taken into account in the calculation of cloud pH.*

The following sentence is added into the text:

"The effect of dust on pH is not taken into account as composition of dusts is not represented within Chimere."

*(6) The volatilities of the three POA components should be mentioned. Also despite the corresponding discussion it is not clear to me how their emissions are calculated from the OA emissions in the corresponding inventory for the various sources. A table with the corresponding volatility-resolved emission rates for the various sources would be helpful.*

The partitioning constant are added into the text. Properties were already shown in Table 3.

The SVOC split is added into the text: "For each sector, emissions of SVOC are splitted into emissions of POAlP (25% of emissions), POAmP (32% of emissions) and POAhP (43% of emissions) to follow the dilution curve of POA in Robinson et al. (2007).

*(7) The POA aging reactions (1)-(3) are a net sink of OH. In the rest of the SOA reactions it is assumed that the OH is recycled. Why are these treated differently?*

It was not treated differently (OH is recycled). The text and the equations are modified.

*(8) If my understanding is correct the authors assume that the primary OA emissions in the inventory that they are using represent 20 percent of the organic compounds emitted in a certain volatility range (less or equal than 100 µg m−3 as IVOCs are neglected). So the corresponding emissions are multiplied by a factor of 5. Given that a lot of these emissions have been measured at high OA levels this increase appears to be too high. It appears that the emissions used set the IVOC emissions to zero and at the same time add a lot of organic emissions to the more volatile SVOCs. This is an important issue for the model so some additional analysis of what exactly is done and why is necessary.*

As discussed in section "Anthropogenic emissions", it was already demonstrated by van der Gon et al. (2015) that biomass burning OA emissions are strongly underestimated. More discussions are added into the text:

"van der Gon et al. (2015) has shown that POA emissions are greatly underestimated due to a strong underestimation of residential wood burning emissions by a factor 3 over Europe (between 1 and 10 depending on the countries) if SVOC emissions are included. This strong underestimation of emissions is due to the use of filter at high temperature (for which SVOC are mainly present as vapors) for emission factor measurement. This result is confirmed by May et al. (2013) who found that 80% of SVOC evaporate at high temperature."

"A sensitivity analysis of the SVOC/POA ratio was already performed by Couvidat et al. (2012)."

Moreover, the importance of SVOC emissions is already discussed in the discussions.

*(9) The use of ISORROPIA-II is rather confusing. Its major advantage compared to ISORROPIA is its ability to simulate the thermodynamics of calcium, magnesium, etc., compounds. However, the authors state in Section 2.1.4 (page 5, lines 22-24) that the crustal elements are not taken into account for the partitioning. This counterintuitive choice requires some justification.*

As written in the text, the composition of dusts is not simulated by the model and therefore composition of dusts cannot be used into ISORROPIA and therefore simple hypothesis were used to

simulate coarse nitrate formation. However, we agree that taking properly dusts into account could be a major improvement.

The following text is added into discussions:

"Interactions of dust with inorganic aerosols could be better represented in the model. The interactions could for example be simulated by taking into account the mineralogy of dusts within Chimere, by emitting dusts with different composition depending on the location of emissions."

*(10) Some discussion of the aerosol size resolution used in the model is needed.*

Added at the beginning of the method section:

"The model uses a sectional approach where particles are separated into several diameter bins. In this study, particles were separated into 10 bins from 10 nm to 10 µm. "

*(11) Section 2.1.8. The way that the mass transfer is between the gas and particulate phases is simulated is not entirely clear. If my understanding is correct, bulk equilibrium is assumed for both inorganics and organics for sizes below 1.25 micrometers. However, it is not clear what happens for coarse particles for compounds like ammonia, hydrochloric acid, organics, etc. Also is the formation of calcium nitrate the only way that nitrate can be transferred to the coarse particles? What about sodium nitrate?*

Explanations are added into the text:

"For particle with a diameter above the cutting diameter, absorption/evaporation is represented by solving the equation of condensation/evaporation (Seinfeld and Pandis, 1998):

$$\frac{dA_{p,i}^{bin}}{dt} = k_i^{bin}(p_i - p_{eq,i}^{bin})$$

with $p_i$ the vapor pressure of i and pbineq;i the vapor pressure of i at equilibrium with the particle-phase concentration of i inside the bin. pbineq;i  is computed with the reverse mode of ISORROPIA for inorganics. However, the condensation of SVOC onto coarse particle is not taken into account."

As detailed in section 2.1.8, the condensation of HNO3 into sea salt leading to the evaporation of HCl is taken into account.

*(12) Tables 1-3. A list of the species names probably in the Supplementary Information is needed to understand the information in these tables.*

We don't think that a list of species would improve the quality of the study. Table 2 provides a list of the main species.

The list of SOA precursors was added to the caption of Table 1. The reference is added to the caption

*(13) The Henry's law constants of a number of organic species are both in Tables 2 and 4 and in quite different units. These important parameters should be in one table with some explanation about their source. Also the fact that they appear to be known quite accurately (three significant digits in Table 2) is problematic.*

The units are harmonized. As said in the text, SOA is formed with the H²O mechanism [Couvidat et al., 2012] and therefore parameters come from H²O. Reference is added to Table 2.

Table 3 presents effective Henry's law parameters (contrary to Table 2). Therefore, values can be different. The following explanation is added to the text:

"For SVOC, Henry's law parameters for wet and dry deposition were computed for a pH of 5.6 (pH of water in presence of $CO_2$)."

*(14) The authors mention that they use the EMEP inventory published in 2003. Are they using any emissions updates for 2013 or the emissions used are that old? A table summarizing the anthropogenic emissions used for each major source category would be helpful together with additional information about the origins of these estimates. The biomass emission rates should be mentioned.*

2003 corresponds to the year of the paper describing the methodology. The EMEP inventory for 2013 have been published in 2015 and provides national official data.

The EMEP inventory is widely used in the modeling of air quality over Europe.

We don't see how it could be possible to summarize emissions and biomass emission rates into a single table as emissions vary spatially with different emissions according to official estimations for each country. Moreover, describing emissions is outside the scope of this paper.

For additional information, please report to the ceip website (http://www.ceip.at/) where all the information of the 2013 inventory can be found.

*(15) How are the sea-salt emissions calculated? What is the assumed composition and size distribution? This scheme appears to perform quite well based on the Chimere evaluation presented here.*

Sea salt emissions are computed as in Chimere 2013 with the parameterization of Monahan. Reference added into the text.

*(16) Some additional discussion of the ammonia emissions used would be helpful. Is there a diurnal variation of the emissions? Is the monthly variation (shown in Fig. 10 for six countries) similar for all areas in the domain? What are the total emissions?*

Total emissions come from the EMEP inventory.

The following precision is added into the text:

"Temporalization of emissions is done according temporal factors for each country provided by GENEMIS [Ebel et al., 1997]."

*(17) A short description of the various stations used in the paper is needed. This could be part of the Supplementary Information. My understanding is that all of them are regional background stations. Is this correct? Are there any exceptions?*

The stations are regional background. Description of the stations are present in the text when 0necessary to the understanding of results. Due to the great number of stations, they will not be described in this publication as lot of information are available in http://www.nilu.no/projects/ccc/sitedescriptions/index.html

*(18) There appear to be around 20 stations measuring sodium in Figure 2. However, 38 stations are mentioned in Table 5. What is causing this discrepancy? There are similar issues with other PM components.*

The number of stations was checked. The right number of stations is plotted in Figures. The figure are modified to make figures more readable.

*(19) A number of urban background stations (e.g., in Paris) could have also been used in the evaluation. Is there a reason why they have been excluded? Also there should be PM1 measurements from the ACTRIS network available for the same period.*

Only open access data available in ebas where used for the comparison. Moreover, due to the huge amount of work that was necessary for this study, not all type of comparison could be done. Moreover, the resolution of the model is not enough (20km) for comparison to urban background stations to be meaningful.

*(20) How well did the measured PM mass concentrations compare to the sum of the components? Could some of the measured PM mass be water as previous studies have indicated for Europe but also for the US?*

Very few stations provide a lot of components. As explained in the text, we try to close the mass balance for station DE0044R (Melplitz) and based on measurements it was not possible to reproduce PM. However, even at this station, more information would be needed on primary components.

The presence of water in measurements is indeed a possibility, but it should depend on the method and the conditions of measurements.

*(21) The number of measurements used in the evaluation should be mentioned in Table 5.*

The number of measurements is added into the text.

*(22) A brief summary of the boundary condition values (or ranges) provided by Mozart for the various sides of the domain would be helpful. It appears that these boundary conditions may be partially responsible for some of the discrepancies between model predictions and observations in areas like Northern Europe.*

Boundary conditions are hourly, vary spatially and are therefore to be summarized in a single range. The impact of boundary conditions are however illustrated in the maps at the limit of the domain.

*(23) The authors discuss the potential errors in the comparison of model-predicted OM values with OC measurements. However, they never discuss the actual values of the OM/OC ratios that they use. I understand that these vary temporally, but the average values for each station for at least the summer and winter could be presented. Given that there are now a lot of measurements of this ratio by High-Resolution Aerosol Mass Spectrometers in locations around Europe a more informed comparison could be made.*

The OM/OC ratios and the method used to estimate those ratios are already discussed in the section "Organic Aerosol". OC is evaluated according to the method of Couvidat et al. (2012) and the uncertainties were already investigated in this study:

"Organic Aerosol concentration measurements are not available in the database. However, measurements for organic carbon (OC) concentrations are available. OC is the mass of carbon inside the organic aerosols. For the comparison, OM/OC ratios (that depend on the composition of organic aerosols, especially the degree of oxidation of compounds) have to be assumed to estimate OC concentrations from modeled OM concentrations. Turpin and Lim (2001) measured the OM/OC ratios at different 25 locations and found ratios between 1.2 and 2.5 and recommended to use a ratio of 2.1 for rural areas. Following Couvidat et al. (2012), OC concentrations were calculated directly from the concentrations of each organic surrogate compounds using their molecular structure to estimate the OM/OC ratio of the surrogate compounds. Several sensitivity tests were conducted by Couvidat et al. (2012) and has shown that the OM/OC simulated by the $H_2O$ mechanism is generally quite low

compared to the OM/OC ratio recommended by Turpin and Lim (2001). An overestimation of OC concentrations by the model could therefore be due to an underestimation of the OM/OC ratio."

OM data were not used for the comparison as they are not freely accessible on the ebase database and were therefore not used in this study. OM and OC refer to very different technique that are not directly comparable. Moreover, OM is often measure in the PM1 fraction and OC in the PM2.5 or PM10 fraction. Based on this, it will be difficult to evaluate the OM/OC ratio.

*(24) The study focuses on daily average measurements. However, some discussion of the diurnal variations of components like OM would have been welcome. There are measurements of these variations in stations like Melpitz that could be useful.*

OM measurements at the Melpitz station are not available for 2013 in the ebas database at the melpitz station and were therefore not used for the comparison in this study.

Moreover, hourly measurements are scarce and the analysis would not be representative.

*(25) The predicted high levels of OC in parts of central and eastern Europe during the winter appear to be too high. I am assuming that these are due to biomass burning. Is this an indication of emissions that are too high? May be atmospheric mixing that is too weak? Some discussion is needed here because other authors have argued that the biomass burning emissions in the European inventory is too low. Could it also be a problem due to the high SVOC emissions used by the model that become particles under low temperatures?*

"During winter, OC is overestimated over a few stations: ES1778R, IT0004R, CH0005R and DE0003R. These 4 stations are under meteorological conditions difficult to simulate at such a low resolution (25 km). They are often close to cities: ES1778R is close to Barcelona (60 km), IT0004R (Ispra) is in the Po valley (an area with strong anthropogenic emissions) not far from Milan, CH0005 is at less than 20 km of Lucerne (with 205 000 inhabitants) and DE0003R is at 12 km from Freiburg (a city of 206 000 inhabitants). Moreover, these stations are in mountainous regions with high variations of altitude that are difficult to represent at such a resolution. Except for these 4 stations, concentrations of organic aerosol in winter simulated by the model tend to be underestimated although reasonable performances could be attained for numerous stations emphasizing the need to better represent anthropogenic emissions in winter."

*(26) There is no discussion regarding the predictions of EC by Chimere.*

As EC measurements are scarce and therefore difficult to generalize, they were not included in the analysis. Moreover, concentrations are low and will not have a significant impact on PM concentrations.

*(27) Some discussion of the evaluation of coarse nitrate predictions by Chimere are needed at least in Cyprus but also in other areas where PM2.5 and PM10 nitrate measurements may be available. A plot of this coarse nitrate in Figure 18 would be useful. Could some of the nitrate prediction problems be due to the challenges in predicting coarse nitrate? (28) The model predicts high nitrate levels in the southeastern Mediterranean (Figure 6). This is rather unexpected due to the relatively low NOx levels and high sulfate concentrations. The values appear to be quite high compared to what has been observed (there are available measurements in Crete). Is this due to dust? How does the model produce so much nitric acid in that area?*

High concentrations of HNO3 are typically simulated over the Mediterranean Sea due to high emissions of NOx and high concentrations of OH. HNO3 can condense onto both the sea salts and the dusts. As shown in section 3.3.3, total concentrations of nitrate is well represented into the model with high concentrations of nitrate measured at the stations whereas concentrations of ammonium are low.

A coarse nitrate evaluation is added on the ammonium and nitrate section.

*(29) Figures 3, 5, 7, 8, and 9 are quite confusing because they contain too much information and do not have legends. One solution would be to show just the predicted and measured concentrations in these figures and then show the evaluation metrics in the supplementary information. It would be nice to also indicate the number of stations in each area in these figures.*

The big interest of these figures is that they summarize all the results of the models and provide a lot of information on monthly and regional performance. Just putting the mean model and the mean observation would oversimplify the figures. The legend was described in the caption. However, these figures were put in the supplementary materials.

The number of stations in each area in now indicate in Table 6.

*(30) The axes of Figure 10 do not have titles.*

The caption was modified:

"Seasonal factors used in CHIMERE to compute the evolution of NH3 emissions for several countries. The factors originate from GENEMIS"

*(31) There are several cases with missing data in the time series of Figures 12, 13, etc. These are replaced by lines connecting the existing measurements. This is confusing and may be misleading. There should be gaps in the corresponding lines. Even better symbols could be used instead of lines for the measurements.*

Measurements were intentionally chosen to be represented as lines. Representing measurements as lines make the figure more readable as it can be very difficult to compare lines to lines. Figures were done to be as readable as possible.

*(32) An explanation of what is shown in the QQ scatter plot (Figure 16) is needed.*

An explanation was already present in the text:

"A Quantile-Quantile (QQ) scatter plot of modeling results against measurements for $PM_{10}$ and $PM_{2.5}$ is shown in Figure 16. QQ plots can be used to assess the similarity of the distribution of two compared datasets."

The following sentence was added to provide more explanation:

"Fig. 16 shows the quantile of modeled concentrations against the corresponding quantile of measured concentrations."

*Other issues*

*(33) The use of Na instead of Na+, SO4 instead of SO2−, etc., throughout the manuscript is problematic and should be avoided.*

Changed.

*(34) There are a number of typos that should be corrected. Some of them: Page 7, line 18. NH3 Page 7, line 26, Henr's Page 11, lines 29-20, with strong observed of Na Page 20, line 24, S. et al. (2016) Page 26, lines 4-5. Missing author names. Figure 3, 5, 7, 8 , 9 captions. monyhly. Page 46, line 1. PM2.5*

Corrected

*Anonymous Referee #2

*The manuscript describes an updated aerosol module for the CHIMERE regional air quality model, along with evaluation of these changes against surface concentration measurements over Europe. The updates cover a number of different processes within the model: emissions, wet deposition, evaporation and condensation of both organic and inorganic semi-volatile components and hygroscopic growth. A set of performance criteria from the literature are adopted and the model is shown to perform well against these. However, while the work itself is a worthy contribution to the field of aerosol modelling for air quality applications, there are a number of deficiencies in the presentation such that I would recommend major revisions before the manuscript is suitable for publication in GMD.*

*General comments*

*1. There are a very large number of figures (21), many of them with multiple panels and similar in nature and with dense high-frequency time series that are hard to interpret. This makes it difficult for the reader to discern what are the important results being presented. If this level of detail is necessary for completeness, it would be better placed in supplementary material, and a smaller number of clearer figures used to (i) exemplify the raw data, and (ii) summarise its meaning statistically in a visual form.*

Some figures were moved to the supplementary materials to simplify the document. These figures already summarize statistically the spatial and temporal evolution of the concentrations.

*2. The manuscript presents an updated version of an existing model; however it is frequently unclear how the new schemes described here compare to those used in the reference/baseline version of CHIMERE (Menut et al., 2013) in their formulation and complexity. Corresponding results for the reference version should also be included, in order to assess not only the absolute performance of the revised model, but to what extent the changes described in this manuscript produce improvements in these performance metrics.*

Addressed in the general reply.

*3. In several places in the manuscript, positive and negative biases and larger or smaller errors and correlations are shown at individual stations and over various regions. These may very well be statistically significant variations, however the analysis presented does not adequately demonstrate this.*

The statistics compute here are based on the results of several stations. Mathematically, several stations will have a larger bias and some a lower bias. When statistics differ significantly from the average statistics over a region, the specific results are discussed in the text. That is why every time, the map of MFB for each station was shown along to the seasonal and regional statistics.

Generally, air quality model evaluation studies present statistics over the whole domain without entering into much details.

*Specific comments*

*4. Page 1, line 1. This describes a "new" aerosol module, although elsewhere it is clear that this is in fact an update to an existing module; the introductory text should be re-worded accordingly.*

It is really a new module has everything has been rewritten (except for the nucleation routine). The text was modified to emphasize on that.

*5. Page 1, lines 1–17. It would be good to see some quantitative results quoted in the abstract about the performance of the updated model and how that compares to the reference/baseline version.*

Addressed in the general reply.

*6. Page 1, line 19–page 2, line 3. The introductory paragraph is quite vague on the subject of why such models are useful, despite the list of model references. A little more background on the motivation for such modelling would be welcome.*

The sentence was modified:

"The development of models is necessary to simulate the formation of particles in order to study processes leading to particle formation, to produce air quality forecasting, to evaluate the efficiency of air pollution mitigation strategies and to study the impact of emissions sources on air quality."

*7. Page 2, line 10. What definition of "fine" is being used in this context?*

PM2.5. Added.

*8. Page 3, line 19–21. An overview of the reference/baseline model version here would be very useful – overall approach and assumptions, what are the tracers used, does it represent the particle size distribution or is it a bulk scheme etc.? This would also make it easier to clarify in the rest of the section how the updated schemes relate to this baseline.*

"The model uses a sectional approach where particles are separated into several diameter bins. In this study, particles were separated into 10 bins from 10 nm to 10 μm."

There is no tracers simulated in this study.

*9. Page 3, line 24–page 4, line 10. How does this compare to the chemical mechanism in the baseline version?*

Addressed in the general reply.

*10. Page 4, lines 13–22. How does this compare to the treatment of biogenic emissions in the baseline version?*

Addressed in the general reply.

*11. Page 4, line 25. A brief discussion of what these emissions are would be helpful, even if further detail is to be found in the reference.*

"VOC emissions (based on the EMEP inventory in this study) are used as in Menut et al. (2013), COV are split into CHIMERE model species according to a speciation database depending on the emission sector."

*12. Page 4, line 25–page 5, line 17. This subsection cites various conflicting studies, but leaves the reader unclear as to what conclusion is drawn for the purposes of this work.*

The subsection was reformulated for further clarity. The confliction works are a justification for why IVOC emissions are not taken into account (contradicting results from various studies).

*13. Page 5, line 20–page 6, line 14. How does this compare to the treatment of aerosol thermodynamics in the baseline version?*

Addressed in the general reply.

*14. Page 6, line 10. A description and/or reference should be provided for the "H 2O mechanism".*

"As in Couvidat et al. (2012)" replaced by "As in the Hydrophilic/Hydrophobic Organic H$^2$O mechanism [Couvidat et al., 2012]"

*15. Page 8, line 1–page 9, line 1. How does this compare to the treatment of wet deposition in the baseline version?*

Addressed in the general reply.

*16. Page 9, lines 3–28. How does this compare to the treatment of condensation/evaporation in the baseline version?*

Addressed in the general reply.

*17. Page 10, lines 1–8. How does this compare to the treatment of coagulation in the baseline version?*

Addressed in the general reply.

*18. Page 10, lines 11–12. There are several IFS-based products from ECMWF. Please clarify whether this refers to operational analyses or forecasts, or to one of one of the reanalyses (e.g. ERA-Interim).*

Operational analyses. Added.

*19. Page 11, lines 7–11. The description of the observations is very brief, and would benefit from being extended – e.g. what type of instruments to these measurements come from, and how extensive is its coverage in space and time? Also, a reference and acronym expansion should be provided for EBAS if possible.*

No acronym was found on the website or the publication.

Reference was already present. It is moved for better presentation.

A few details were added:
"Results of the model are compared to various measurements (NO3-, NH4+, SO2, Na+, Cl-, OC, PM1, PM2:5 and PM10) available in the EBAS database (Tørseth et al., 2012) from various instruments (i.e. filters, Tapered Element Oscillating Microbalances, beta ray absorption) for regional background stations. The stations cover most of Europe with the first measurements available beginning in the seventies. EBAS is a database hosting observation data of atmospheric chemical composition and physical properties in support of a number of national and international programs ranging from monitoring activities to research projects. EBAS is developed and operated by the Norwegian Institute for Air Research (NILU). This database is mostly populated by the EMEP (European Monitoring and Evaluation Program) measurements"

*20. Page 11, lines 23–24. Please explain why this is a likely explanation for the Na and Cl results.*

This sentence was removed as this result is not clear enough.

*21. Page 13, lines 23–25. It could also be that a third factor which is poorly captured in the model affects both sulfates and nitrates.*

Indeed nitrate could be a source of error too. The following sentence was added in the text:

"Part of the errors may be also due to errors on $NO_3^-$ and $HNO_3$ concentrations."

*22. Page 14, line 20. Please describe and/or give a reference for MELCHIOR 2.*

A reference for Melchior 2 was added.

*23. Page 15, line 9. It is not clear whether "OC concentrations were calculated directly" from the model or from observations. Please clarify.*

Clarified: "modeled OC concentrations were calculated directly from the modeled concentrations of each organic surrogate compounds"

*24. Page 16, line 6. MFB for PM10 is still positive, suggesting coarse particles are still overestimated, just less so than smaller particles.*

No coarse particles are underestimated but PM10 remains overestimated due to the overestimation of PM2.5. We modify the text to compare model PM coarse concentrations directly to the differences between PM10 and PM2.5.

*25. Page 17, line 26. This sounds like the measurements are overestimated, but presumably is intended to say that the model overestimates NO3 compared to the measurements?*

Corrected. "Measured" was used instead of "modeled"

*26. Page 17, line 34–page 18, line 1. Please explain why a lack of HNO3 condensation is likely to explain this.*

An explanation was added: "with more $HNO_3$ condensing onto dust and sea salt, less $HNO_3$ will be available to form ammonium nitrate"

*27. Page 20, line 23. A reference to the data referred to here would be good.*

A reference was added.

*Technical corrections*

*28. Page 1, line 8 (and elsewhere): Performances were −→ Performance was. 29. Page 1, lines 8 and 10 (and elsewhere): sea salts −→ sea salt. 30. Page 1, line 15: most of stations −→ most of the stations.*

Corrected

*31. Page 2, line 18: dusts −→ dust.*

Corrected

*32. Page 2, line 24: aerosols thermodynamics −→ aerosol thermodynamics.*

Corrected

*33. Page 3, line 15: described in which part of the paper?*

Corrected

*34. Page 3, lines 25–26: "a" should be before "function", not "partitioning".*

Corrected

*35. Page 3, line 28: insert "and" before "equilibrium constants".*

Corrected

*36. Page 4, line 14: "temperature" and "solar" should not be capitalised.*

Corrected

*37. Page 4, lines 16–17: "wilting point" should not be capitalised.*

Corrected

*38. Page 4, line 20: landuse −→ land-use.*

Corrected

*39. Page 5, line 14: vehicle −→ vehicles.*

Corrected

*40. Page 5, line 15: aromatics compounds −→ aromatic compounds.*

Corrected

*41. Page 9, line 13: Delta −→ Δ.*

Corrected

*42. Page 9, lines 15 and 17: what is Kn?*

The Knudsen number. Description added.

*43. Page 9, line 21: insert "in" before "computed with".*

Sentence corrected.

*44. Page 9, line 25: CaCO3 of dusts −→ CaCO3 in dust.*

Corrected

*45. Page 10, lines 12 and 15: "temperature" should not be capitalised.*

Corrected

*46. Page 10, line 15: "wind speed" should not be capitalised.*

Corrected

*47. Page 10, line 16: underestimation of the PBL around. . . −→ underestimation of the PBL height of around. . .*

Corrected

*48. Page 10, lines 17–18: "boundary conditions" should not be capitalised.*

Corrected

*49. Page 10, lines 21–22 (and elsewhere): performances −→ performance.*

Corrected

*50. Page 11, line 25: than −→ as.*

Corrected

*51. Page 11, lines 29–30: observed of Na −→ observed Na.*

Corrected

*52. Page 14, line 3: "summer" should not be capitalised.*

Corrected

*53. Page 15, line 19: criteria is −→ criteria are.*

Corrected

*54. Page 15, lines 19–20: overestimation. . . do not correspond −→ overestimation. . . does not correspond.*

Corrected

*55. Page 16, line 18: delete "the" before "Southern Europe".*

Corrected

*56. Page 16, line 24: PM2.5 are −→ PM2.5 is.*

Corrected

*57. Page 16, lines 25, 31–32: "winter", "spring", "summer" and "fall' should not be capitalised. Also, please use either "fall" or "autumn" consistently throughout – both appear in the manuscript.*

Corrected

*58. Page 16, line 32: due to mostly to −→ due mostly to.*

Corrected.

*59. Page 17, lines 20–23. This sentence is confusing, with two consecutive "but" clauses and multiple parentheses. Consider breaking it up to clarify the meaning.*

Corrected

*60. Page 17, line 27: faction −→ fraction.*

Corrected

*61. Page 18, line 14: "April" and "August" should be capitalised.*

Corrected

*62. Page 18, line 31: could be explain −→ could be explained.*

Corrected

*63. Page 19, line 4: emissions is −→ emissions are.*

Corrected

*64. Page 20, line 13: inorganics aerosol −→ inorganic aerosol.*

Corrected

*65. Page 20, line 24: the "S. et al" citation should have a full surname, not just an initial. (The same applies to all authors in the corresponding bibliography entry.)*

Corrected.

*66. Page 21, line 9: dynamic −→ dynamics.*

Corrected

*67. Page 21, line 14: "CHIMERE" should be capitalised as elsewhere in the manuscript.*

Corrected

*Anonymous Referee #3*

*This manuscript presents an update of the CHIMERE's aerosol module. The chemical mechanism was modified for the formation of the secondary organic aerosol precursors. The equilibrium between the aerosol and the gas phase is then treated using the module SOAP. For the secondary inorganic aerosols, the thermodynamic equilibrium is computed using ISORROPIA, which has been updated to the version 2.1 here. Biogenic emissions have been updated and are now computed using MEGAN 2.1. Belowcloud scavenging has also been updated. After a presentation of the model CHIMERE and the aerosol module, the developments are validated over the year 2013 using surface measurements from the EBAS database by separating Europe into 5 coherent sub-regions based on country borders. The authors give some recommendations for paper future development. Finally the manuscript ends with a conclusion. This manuscript is interesting for the aerosol community as it presents recent features in aerosol modelling. However, it presents some serious issues for a publication as it is. The manuscript presents a new aerosol module (cf. abstract), but otherwise in the text it is treated as an update of the existing module. It is sometimes difficult to differentiate the parametrization and developments related to the 2013 version from the new 2017 version. The authors need to be clearer about this point that highlights the important development work done here. I would recommend to present the 2013 version of the parametrizations before introducing the new features. Following this remark, there is no comparison between the 2013 and 2017 versions over the year 2013. It would be very interesting to see the evolution in the performance of the model between the two versions.*

It is indeed a new module. This point is addressed in the general comments.

*The comparison to the observation set is very interesting as it uses a lot of collocated information on several measuring stations. It is then possible to evaluate the aerosol load, but also the aerosol composition and the seasonality. All these pieces of information could point out easily the strengths and the weaknesses of the model. However the authors only use ground based stations. It would have been interesting to compare the simulation to vertically integrated measurements such as aerosol optical depths, especially to evaluate the impact of the changes on wet deposition.*

Ongoing works are carried out on deposition which will have its specific evaluation. Indeed, comparing to aerosol optical depths present many interests and coupling this to the analysis of wet deposition

could be very useful. However, it was chosen in this study to concentrate on ground based stations (because of the huge amount of work) to evaluate the capacity of the model to simulate composition of particle.

*Also, almost all the mathematical formulations need to be reviewed. There are for example undefined variables used in equations or discrepancies between the name of a variable in the text and in an equation. I would then recommend major revisions before publication.*

Equations were revised.

*General comments:*

*1. The use of paragraph breaks is sometimes puzzling, for example on page 13. This sentence might make think that the next paragraph does not refer to the figures 7 and 8, which is not the case.*

Corrected.

*2. Page 2 line 2: I do not understand the presence of the "Vestreng, 2003" reference for the air quality model.*

Bug inside bibtex. Corrected.

*3. Section 2.1.1: Maybe a table that summarizes the reaction rate for the oxidation of SO2 in clouds could be useful.*

The reactions were added in the text.

*4. Section 2.1.3: I didn't understand how are the anthropogenic emissions managed for the POA and SVOC. You use POA emissions from an emission inventory. These POA are emitted into the species POAlP, POAmP and POAhP. Then you use the quantity of POA emitted to emit the SVOC by saying that SVOC = 5 \* POA and say that you don't take into account the IVOC. Is this right? How are then the SVOC emitted into the MELCHIOR species?*

POA are transformed into SVOC and the SVOC are splitted into POAlP, POAmP and POAhP. POA emissions are transformed into POAlP, POAmP, POAhP emissions. The text is modified to improve clarity:

"In this study, POA are transformed into SVOC emissions: a SVOC/POA of 5 is used for residential emissions (without adding IVOC emissions and assuming that POA emissions only account for 20% of emissions) and 1 for the other sectors (assuming therefore that no SVOC emissions from other sectors are missing).

For each sector, emissions of SVOC are splitted into emissions of POAlP (25% of emissions), POAmP (32% of emissions) and POAhP (43% of emissions) to follow the dilution curve of POA in Robinson et al., 2007."

These species do not correspond to MELCHIOR species. They are added and are assumed to not impact OH radical and therefore the gas chemistry. The text is modified to emphasize on that.

*5. Page 6 line 4: I understand the dynamic method requires a lot of computation time explaining you choices. But did you run a test to know the impact it could have on a specific test-case for example?*

This is undergoing work that should be published soon. This particular method requires a lot of CPU time (increase of a factor 10) and the effect of such a method on concentrations is a study in itself.

*6. Page 7, line 1: pleas also add "the mass fraction of respectively the solid phase, the aqueous phase and the organic phase in the particle" for a better understanding.*

Added.

*7. Section 2.1.6: The title is "Dry deposition of particles and semi-volatile organic species" but you talk about other gaseous species such as O3, SO2, etc. Please change the text to be consistent.*

Corrected.

*8. Section 2.2: When talking about the simulation set-up, you don't talk about the vertical resolution of the simulation made. Please add these informations.*

Corrected.

*9. Page 10, line 18: You wrote "Boundary Conditions were generated from [...]". Could you explain how they were generated?*

The text is modified:

"Boundary conditions were generated from the results of the Model for OZone And Related Tracers (MOZART v4.0) available online on https://www.acom.ucar.edu/wrf-chem/mozart.shtml."

*10. Section 2.3: This part is too small. Could you please add a table with the total number of stations and the number of stations in each of the regions you defined. According to Fig. 1 there are no stations for western Europe. Could you please explain why? Also, please add the link to the EBAS database in the text.*

A link for EBAS and some information were added.

Fig 1. Shows only the stations that are referred in the text. These stations are mostly stations with measurements of OC. No particular station in western Europe was discussed in text (mostly because there are no OC measurements).

The number of stations in each area in now indicated in Table 6.

*11. Page 11, line 19: You wrote "Only one station in Spain underestimates [...] concentrations for Na." while on Fig. 1 there are three blue triangles.*

Corrected. "Concentrations for Na$^+$ are underestimated significantly for only one station in Spain (along the Bay of Biscay)"

*12. Page 11, line 29: You talk about the station ES0008R, but without showing any figures. Also this sentence is hard to understand.*

The sentence was reformulating. A figure was added in supplementary Materials.

*13. Page 12 line 2: "The other stations [...] the opposite trend". I do not understand which other stations? All the stations or the other stations from southern Europe? Could you please make this part clearer?*

The sentence was reformulated:

"Except from ES0008R, the stations of Southern Europe share the same pattern shown"

*14. Section 3.2: You are talking about sulfate observations. But are you using "total sulfate" or "corrected sulfate" (or non sea salt sulfate) measurements?*

We use non-corrected measurements (so the "true" sulfate measurements) as sea salt sulfate is taken into account in the model.

*15. Page 14, line 23: I do not understand how an overestimation of NH3 emissions could induce an overestimation of TNO3. Could you please explain it with more details?*

"because the deposition velocities of these gases are generally higher than those of particles" was added to the explanation.

*16. Section 3.3.1: How many measurements do you have for each stations in Table 6? it seems that some stations have very few measurements points, e.g. CH0002R looks like to have around 10 points. What is the confidence in the statistics you can have in this case?*

Statistics are weighted by the number of measurements points. A station with a low number of point will a low weight and therefore has a low impact on the statistics.

*17. Page 16, line 16: "especially in the Alps". How is the relief represented in the model in this area? Can it explain the overestimation of the PM by the model? Maybe you could try to interpolate the model at the altitude of the stations to improve the comparison made.*

The model doesn't really take into account the relief. That is a classic problem for 3D air quality models that runs at such a coarse resolution and yes it should be one of the main reason explaining the overestimation.

We try to interpolate according to the altitude but it didn't really improve the results. The best way to handle this problem would be run the model with a high resolution.

The following sentence was added to the text:

"The overestimation over the Alps is probably due to difficulties in reproducing the complexity of

mountainous meteorology for a model with such a coarse resolution."

*18. Page, line 30: What is the effect of the strong overestimation over the Alps on the results of Fig. 15 and 17 for central Europe?*

The effect is quite low. For example, we removed the 3 stations for PM2.5 over Alp that are overestimated. The results are illustrated below. The RMSE and the MFB are a bit lower but the results are similar.

Without the 3 stations

[Figure]

With the stations

[Figure]

*19. Section 3.3.3: You talk about Ca and NO3 in PM10, but in Fig. 18 there is "total Ca" and "total NO3". Are these two quantities supposed to be the same? Please change the text to be clearer.*

Total replaced by "in PM10"

*Figure related comments:*

*1. The manuscript preparation guidelines for authors claims that: The abbreviation "Fig." should be used when it appears in running text and should be followed by a number unless it comes at the beginning of a sentence, e.g.: "The results are depicted in Fig. 5. Figure 9 reveals that...". Please check that the right word is used in the text.*

Corrected.

*2. Also the authors should check the legend of their figures that are not always satisfactory. For example, the legend of the first figure does not mention what are the measuring station marked by dots.*

Corrected.

*3. Concerning the time series, it seems that there are missing values in the observations linked by a segment in the figures (e.g. On Fig. 12 between Feb and Mar 2013 for GR0002R). This can be misleading for the interpretation of the figure. Could you please change this? Maybe you could use symbol for the measurements and have continuous line for the simulation results.*

Measurements were intentionally chosen to be represented as lines. Representing measurements as lines make the figure more readable as it can be very difficult to compare lines to plots. Figures were done to be as readable as possible.

*4. Figure 2, 4, 6, 11 and 14: These figures does not seem to be complete on the right side for negative numbers. What represent the squares on these figures?*

Corrected.

Squares were modified by triangles.

*5. Figure 3, 5, 7, 8, 9, 15, 17: Is it possible to have slightly thicker lines? Is it possible to draw a line for 0 in order to read more easily some quantities such as the MFB.*

Based on the other reviews, these figures were moved to the Supplementary because they are quite difficult to read.

*6. Figure 3: All your sub figures does not have the same size or position.*

Corrected

*7. Figure 12 and 13: All your sub figures does not have the same size.*

Corrected

*8. Figure 13: There seems to be a mistake. I think "CH0005R" should be "CY0002R".*

The Cyprus station was missing. Corrected.

*9. Figure 16: The figure for PM2.5 is very small. Is it possible to enlarge it?*

Changed.

*Table related comments:*

*1. Table 2, line "BiDer": hydrohilic -> hydrophilic.*

Corrected.

*2. Table 2: for the type you mention type A, B and C, but in the table you only write hydrophilic or hydrophobic. Please explain this a little more.*

Removed.

*3. Table 4: What are the Henry's law constant used for the other species such as HNO3?*

"High numerical values were used for HNO₃ and for gaseous H₂SO₄ to take into account their hydrophilic properties." Was added in the caption of the table.

*4. Page 14, line 8: Please add a reference to Table 5 when talking about TNO3 and TNH4.*

Added

*5. Table 5 presents results for O3 and NO2 but you never talk about them in the text.*

Removed from the table.

*Technical comments (when a letter or a word is missing, it is in bold in the comment):*

*1. Page 3, line 15: I guess the word "first" is missing in the sentence.*

Corrected.

*2. Page 5, line 19: "Thermodynamic of Secondary organic and [...]" -> "Thermodynamic of secondary organic and [...]"*

Corrected.

*3. Section 2.1.5: the symbols w are different but refers to the same quantity. So are d and d. There is also a discrepancy between dsol and dsolid.*

Corrected.

*4. Page 7, line 9-12: Fd, i is different from Fdry,i. Same for vd,i, vd and vd, i. What represents Ci?*

Corrected.

*5. Page 7, line 25: ρeau and ρ. What is Meau? Does Hi stand for the Henry's law constant?*

Corrected.

*6. Page 7, line 26: Henry's law.*

Corrected

*7. Page 9, line 12: there is no verb in the sentence.*

Corrected

*8. Page 9, Eq. 18: What is P?*

Added. Precipitation rate in mm/h.

*9. Page 9, Eq. 19: Delta -> Δ, k bin -> k bin i*

*Corrected*

*10. page 9, Eq. 20: What is R and T?*

Added

*11. Page 9, Eq. 21: Delta -> Δ*

Corrected

*12. Page 10, line 2: J bin coag,i and J b coag,i*

Corrected

*13. page 10, Eq. 23: Kj,k instead of Kj,l. What is Ap,i? The second sum symbol does not have attribute. Is the sum going from 1 to b over j?.*

Corrected

*14. Page 10, line 13: Wind Speed -> wind speed (also line 15).*

Corrected

*15. Page 10, line 13: Planetary Boundary Layer (PBL) -> planetary boundary layer (PBL).*

Corrected.

*16. Page 10, line 15: Temperature -> temperature.*

Corrected

*17. Page 10, line 18: Mozart -> MOZART.*

Corrected.

*18. Page 10, line 18: "Boundary Conditions" -> "boundary conditions"*

Corrected

*19. Page 11, line 5: Figure 1 -> Fig. 1.*

Corrected.

*20. Page 12 line 17: Tsyro et al. (2011) and Neumann et al. (2016).*

Corrected.

*21. Page 15, line 11:"OM/OC simulated" -> "OM/OC ratio simulated".*

Corrected

*22. Page 15, line 29: "summer concentrations are underestimated in summer". Summer is written twice.*

Corrected

*23. Page 16, line 18: "PM2.5 and PM10, respectively" -> "PM2.5 and PM10 respectively", no comma.*

Corrected

*24. Page 18, line 9: "Figure 12" -> "Fig. 13".*

Corrected.

*25. Page 19, line 9: "to take into IVOC emissions", it seems that a word is missing here.*

Corrected.

---

## Author Comment (AC2) · 26 Oct 2017

About the code availability, this version of Chimere is currently available on request because of the Chimere development policy. The internal rules concerning major model developments is as follows: (1) improve model processes in the framework of research projects (2) publish the scientific results and make the beta model version available upon request (3) After publication of the article and validation by the peer-reviewed process, the publication of the code on the publicly available website could be considered (4) update the forecast version with this latter updated and validated version

2017.